# Viral modulation of type II interferon increases T cell adhesion and virus spread

Carina Jacobsen [1], Nina Plückebaum [1], George Ssebyatika [1,2], Sarah Beyer[1], Lucas Mendes-Monteiro [1], Jiayi Wang [1], Kai A. Kropp[1], Víctor González-Motos[1,3], Lars Steinbrück [1], Birgit Ritter[1], Claudio Rodríguez-González[4,5,6], Heike Böning[1], Eirini Nikolouli[4,6], Paul R. Kinchington [7], Nico Lachmann [4,5,6,8], Daniel P. Depledge [1,6,9], Thomas Krey[1,2,6,10,11] & Abel Viejo-Borbolla [1,6] ✉

During primary varicella zoster virus (VZV) infection, infected lymphocytes drive primary viremia, causing systemic dissemination throughout the host, including the skin. This results in cytokine expression, including interferons (IFNs), which partly limit infection. VZV also spreads from skin keratinocytes to lymphocytes prior to secondary viremia. It is not clear how VZV achieves this while evading the cytokine response. Here, we show that VZV glycoprotein C (gC) binds IFN-γ and modifies its activity, increasing the expression of a subset of IFN-stimulated genes (ISGs), including intercellular adhesion molecule 1 (*ICAM1*), chemokines and immunomodulatory genes. The higher ICAM1 protein level at the plasma membrane of keratinocytes facilitates lymphocyte function-associated antigen 1-dependent T cell adhesion and expression of gC during infection increases VZV spread to peripheral blood mononuclear cells. This constitutes the discovery of a strategy to modulate IFN-γ activity, upregulating a subset of ISGs, promoting enhanced lymphocyte adhesion and virus spread.

Inhalation of varicella zoster virus (VZV) in naïve persons results in infection of epithelial cells of the respiratory mucosa and lymphoid organs of the Waldeyer's tonsillar ring. The close interaction with immune cells, including T cells, results in their infection and systemic VZV dissemination[1–3]. VZV modifies the receptor expression profile of infected T cells, increasing the level of proteins that facilitate T cell homing to the skin and basal stem niches of hair follicles[4], where infectious viruses spread to keratinocytes[5]. A slow infection partly controlled by innate responses ultimately leads to the eruption of the virus at the differentiating surface epithelia, resulting in the typical chickenpox rash. At later stages of infection, another wave of migrating mononuclear cells reaches the skin, and VZV spreads from keratinocytes to lymphocytes, causing secondary viremia[5,6]. How VZV spreads from epithelial cells to lymphocytes is not well understood.

VZV is human-specific, and animal models do not fully reflect VZV pathogenesis. However, the use of severe combined immunodeficiency (SCID) mice xenografted with human skin and dorsal root ganglia supports the relevance of T cell migration for VZV spread and

[1]Institute of Virology, Hannover Medical School, Hannover 30625, Germany. [2]Institute of Biochemistry, University of Lübeck, Lübeck 23562, Germany. [3]University of Veterinary Medicine Hannover, Foundation, Hannover 30559, Germany. [4]Department for Pediatric Pneumology, Allergology and Neonatology, Hannover Medical School, Hannover 30625, Germany. [5]Biomedical Research in Endstage and Obstructive Lung Disease Hannover (BREATH), German Center for Lung Research (DZL), Hannover, Germany. [6]Cluster of Excellence RESIST (EXC 2155), Hannover Medical School, Carl-Neuberg-Straße 1, 30625 Hannover, Germany. [7]Departments of Ophthalmology and of Molecular Microbiology and Genetics, University of Pittsburgh, Pittsburgh, PA, USA. [8]Fraunhofer Institute for Toxicology and Experimental Medicine ITEM, Nikolai-Fuchs-Str. 1, 30625 Hannover, Germany. [9]German, Center for Infection Research (DZIF), Hannover, Germany. [10]Centre for Structural Systems Biology (CSSB), 22607 Hamburg, Germany. [11]German Center for Infection Research (DZIF), Partner Site Hamburg-Lübeck-Borstel-Riems, 22607 Hamburg, Germany. ✉e-mail: viejo-borbolla.abel@mh-hannover.de

pathogenesis in vivo[2,5,7]. Lymphocyte migration is a complex process that requires the concerted action of different proteins including chemokines, adhesion molecules and integrins. Chemokine interaction with their receptors on the T cell leads to activation of integrins, such as lymphocyte function-associated antigen 1 (LFA-1)[8]. Similarly, chemokines and interferon-gamma (IFN-γ) increase the expression of intercellular adhesion molecule 1 (ICAM1) on the endothelium and epithelium[9]. The interaction between LFA-1 and ICAM1 facilitates firm T cell adhesion and transmigration[9,10], the formation of the immunological synapse[11], and the cytotoxic T cell response that kills infected cells[12,13].

The innate response to viral infections includes the expression of type I, II and III IFNs, key antiviral cytokines that bind specific receptors to induce the expression of hundreds of IFN-stimulated genes (ISGs). The expressed ISGs and their level of expression are characteristic for each IFN, and even vary for the same IFN in different cell types[14]. Interestingly, the mode of binding of IFN-γ to the IFN-γ receptor (IFNGR) also influences the induction of ISGs, as seen with recombinant IFNGR agonists that induce biased signalling and differential expression of ISG subsets[15]. One of the differentially expressed ISGs upon binding of a biased agonist is *ICAM1*[15].

Due to the role of chemokines and IFNs in the antiviral response, viruses have devised many strategies to modulate their activities. Herpesviruses and poxviruses express viral chemokine binding proteins (vCKBP) that bind and modulate chemokine function[16,17]. Most vCKBP discovered to date inhibit chemokine function with the exception of herpes simplex virus (HSV) glycoprotein G and VZV glycoprotein C (gC), that enhance chemokine-mediated migration of leucocytes[16–20]. VZV gC is not required for growth in cell culture but is important in skin infection[21]. Until now, viral proteins that bind soluble

IFN have only been discovered in poxviruses[22,23]. These IFN-binding proteins bind IFN with high affinity and compete with the interaction with their receptor, inhibiting IFN activity, and their deletion or mutation severely attenuates the virus in vivo[24–26].

Since gC increases T cell chemotaxis[18], a process influenced by chemokines and IFNs, and is relevant for efficient spread in human skin[5,21], we sought to investigate whether gC could modulate cytokine activity and spread from epithelial to T cells. We show here that VZV gC binds type II IFN with high affinity. Contrary to what has been observed for poxviruses, VZV gC did not inhibit IFNGR signalling and induction of ISGs. Interestingly, VZV gC binding resulted in a biased activation of IFN-γ-mediated stimulation that increased the expression of a subset of ISGs, including *ICAM1*. Increased levels of ICAM1 at the plasma membrane facilitated the adhesion of T cells expressing LFA-1. We also observed more efficient VZV spread from epithelial HaCaT cells to Jurkat cells and to peripheral blood mononuclear cells (PBMCs) when gC was expressed during infection. Collectively, we report a previously undescribed activity of viral modulation of IFN-γ that results in the induction of biased ISG expression, increasing ICAM1 levels, T cell adhesion and virus spread.

## Results

### VZV gC binds to type II IFN

VZV gC is an important virulence factor in in vivo infections of human skin in SCID mice[21] that enhances chemokine-dependent migration[18]. Due to the relevance of IFNs in antiviral responses and lymphocyte adhesion, we performed a surface plasmon resonance (SPR) binding screening with human IFNs. VZV gC is a type I transmembrane protein with an ectodomain (ECD) containing an N-terminal repeated domain (termed R2D) and a larger C-terminal region (formerly termed immunoglobulin-like domain (IgD))[18], a transmembrane region, and a very short cytoplasmic tail (Fig. 1a). We used Phyre2 (http://www.sbg. bio.ic.ac.uk/~phyre2/html/page.cgi?id=index) server predictions of the secondary structure of the gC ECD to design truncated gC expression constructs. We expressed in insect cells, purified (Supplementary Fig. 1, uncropped blots in Supplementary Fig. 17), and immobilised gC constructs onto CM5 Biacore chips. We also expressed and purified $gC_{S147-V531}$ in HEK293ExPi cells to control for differential glycosylation between insect and mammalian cells. The expression of mammalian (m) $mgC_{S147-V531}$ resulted in low concentration (Supplementary Fig. 1i), precluding immobilisation on Biacore chips. $gC_{P23-V531}$, corresponding to the full-length ECD, bound IFN-β, IFN-ω, IFN-γ, IFN-λ1 and IFN-λ2 but did not bind IFN-α and tumour necrosis factor-alpha (TNF-α) (Fig. 1b). To identify the region required for interaction with IFN, we immobilised $gC_{S147-V531}$, $gC_{Y322-S523}$ and $gC_{Y419-S523}$ on a CM5 chip. All three truncations of the gC ECD bound to the different IFNs, except IFN-α subtypes (Supplementary Fig. 2a). To confirm the gC–IFN interactions, we used grating-coupled interferometry (GCI) to perform repeated analyte pulses of increasing duration (RAPID) experiments. We immobilised $gC_{S147-V531}$, $gC_{Y322-S523}$ and $gC_{Y419-S523}$ on a DXH chip and injected IFN-β, IFN-γ, IFN-λ1, IFN-λ2, IFN-ω and TNF-α as negative control. Injection of IFN-γ on chips immobilised with $gC_{S147-V531}$ and $gC_{Y322-S523}$ led to a high response, while the response for $gC_{Y419-S523}$ was very low, suggesting a low-affinity interaction or no binding (Supplementary Fig. 2b, top row). Very low or even no responses were observed for the other tested IFNs against the different gC constructs, suggesting that gC-bound type I and III IFNs weaker than type II IFN (Supplementary Fig. 2b, notice the different scales in the Y axis).

Overall, our results suggest that only the gC–IFN-γ interaction may be of functional relevance.

### VZV gC induces biased expression of IFN-γ stimulated genes

We employed an unbiased approach to determine the functional impact of gC on IFN-γ by analysing the transcriptome of HaCaT, a keratinocyte cell line, 4 h after incubation with $gC_{S147-V531}$ and IFN-γ,

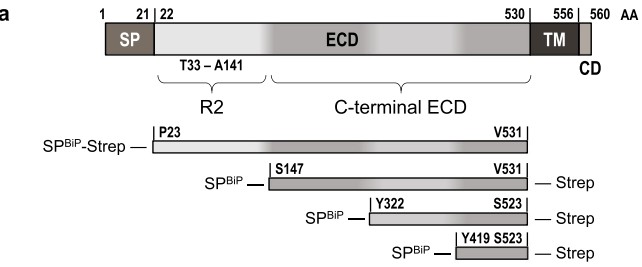

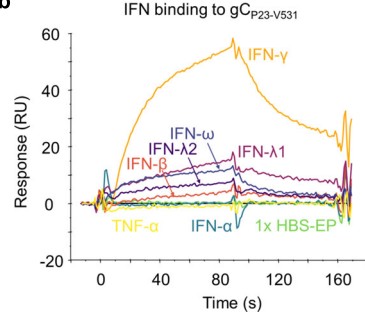

**Fig. 1 | VZV gC binds type II IFN. a** Schematic representation of VZV gC (top) and recombinant soluble gC constructs that were used in this study (below). All constructs contain a BiP signal peptide at the N-terminus and a Twin-Strep-tag. The first and last gC residue is indicated in each construct. The numbering of amino acid residues corresponds to the sequence of the Dumas strain. AA amino acid, SP signal peptide, ECD extracellular domain, TM transmembrane domain, CD cytoplasmic domain, R2 repeated domain 2, BiP Drosophila immunoglobulin binding chaperone protein signal sequence, Strep Twin-Strep-tag with enterokinase site for optional removal of tag. **b** Sensorgram showing the results of a binding screening between VZV gC and cytokines using the Biacore X100 system. The recombinant purified VZV $gC_{P23-V531}$ was immobilised on a CM5 chip (3600 RU). IFNs and TNFα were injected at 100 nM with a flow rate of 10 μL/min. One representative experiment out of three biological repetitions is shown. s seconds, RU resonance units.

both alone and in combination. The principal component analysis (PCA) showed that the different experimental groups separated based on IFN-γ treatment (Supplementary Fig. 3a).

We determined RNA fold changes by comparing the different groups and their significances using DESeq2 (Source Data). Genes were considered significantly differentially expressed between the experimental conditions if they had an adjusted $P$ value lower than 0.05 and a log2-fold change greater than 0.58 (equivalent to a 1.5-fold change). The addition of IFN-γ significantly modulated the expression of 776 genes in HaCaT cells (Supplementary Fig. 3b). The combination of $gC_{S147-V531}$ and IFN-γ raised the number of significantly regulated genes to 831 (Supplementary Fig. 3c), while $gC_{S147-V531}$ alone induced significant expression of just 16 genes (Supplementary Fig. 3d), among them chemokines, signalling molecules and the adhesion molecule $ICAM1$. When comparing the impact of adding $gC_{S147-V531}$ and IFN-γ together versus IFN-γ alone, we observed a significant increase in the expression of 28 genes and 1 pseudogene (Supplementary Fig. 3e).

We generated a heatmap showing the 42 genes with a significant expression change after the addition of gC (Fig. 2a, b). Dendrograms showed that, in line with the PCA plot (Supplementary Fig. 3a), treatment conditions clustered appropriately (Fig. 2b). The majority of changes induced by IFN-γ, gC or both resulted in higher gene expression compared to the mock control (Supplementary Fig. 4a). We performed functional enrichment analysis with Reactome (v86) to identify the biological pathways that were modified upon incubation with both gC and IFN-γ. The most enriched pathways were IL-10 signalling, chemokines and their receptors (Supplementary Fig. 4b). When comparing the effect of 'both vs. IFN-γ', we did not observe significantly downregulated genes. Next, we plotted the log2-fold changes of the significantly regulated genes against each other (Supplementary Fig. 4c). We defined a corridor (grey lines) in which the fold change was less than 1.5-fold. Genes outside this corridor were significantly differentially regulated by gC. A distinct set of genes, including chemokines ($CXCL10$ and $CXCL11$), pro-inflammatory cytokines ($IL6$ and $IL32$), the E3 ubiquitin ligase $NEURL3$[27] and also $IL4I1$, an enzyme involved in immunosuppression[28], as well as a pseudogene ($OR2l1P$), were strongly regulated upon the combined treatment, but not when gC was added alone.

The log2-fold change is only a relative value and does not depict absolute changes that could largely differ depending on the baseline gene expression in the presence or absence of IFN-γ. Therefore, we calculated and plotted the effect size for the 42 significantly regulated genes by gC (Fig. 2c–f, Source Data). In addition to the four major groups observed in the heatmap (Fig. 2b), we divided the genes into two categories, those that were regulated by IFN-γ in our datasets and those that were not. Among the genes that were not regulated by IFN-γ, the effect size of $IL4I1$ was 7.4-fold higher in the gC plus IFN-γ condition than with gC alone (Fig. 2d). The effect on $IL4I1$ expression seen in the 'gC vs. mock' comparison was not statistically significant, while it was when comparing 'both vs. IFN-γ'. This result, together with the fact that IFN-γ did not upregulate $IL4I1$ and due to its role in immune modulation[28], highlights $IL4I1$ as an interesting target for further studies. Additionally, among the genes not regulated by IFN-γ, we observed seven genes with more than a 1.5-fold difference in their effect sizes, when comparing the effect size of gC in the presence or absence of IFN-γ. These include chemokines and transcriptional regulators: $CCL20$, $CITED4$, $CXCL2$, $CXCL3$, $CXCL8$, $DUSP5$ and $HIVEP2$ (Fig. 2d–f).

Interestingly, the picture was different when looking at the genes regulated by IFN-γ: 82% of the genes showed differences for the two comparisons ('gC vs. mock' and 'both vs. IFN-γ', Fig. 2c, d, f). The co-stimulation with gC and IFN-γ upregulated the expression of eleven genes and one pseudogene. Intriguingly, five of these gene products are involved in T cell migration: ICAM1 and the chemokines CCL2, CXCL9, CXCL10 and CXCL11. For $ICAM1$, gC led to about 26-fold higher

effect size in the presence of IFN-γ compared to the condition without IFN-γ.

Overall, the transcriptomic analysis shows that $gC_{S147-V531}$ alone induced the expression of few genes, some of them also regulated by IFN-γ while others were not. $gC_{S147-V531}$ did not induce a general enhancement of IFN-γ-stimulated genes but increased the expression of a subset of specific genes, especially those involved in chemokine-mediated migration and adhesion. This suggests that gC induces a biased expression of ISGs.

## VZV gC modifies the activity of IFN-γ, leading to higher expression of ICAM1

Since gC is a known vCKBP that modulates chemotaxis, we focused mainly on the genes involved in migration and cell adhesion. We confirmed by RT-qPCR the increased expression of $CXCL8$, $CXCL9$, $CXCL10$, $CXCL11$, $IL4I1$ and $ICAM1$ in the presence of gC and IFN-γ (Supplementary Figs. 5, 8). The $gC_{Y419-S523}$ construct that bound IFN-γ less efficiently did not increase IFN-γ-dependent gene expression as efficiently as $gC_{S147-V531}$ (Supplementary Fig. 6). The IFN-γ employed in the previous experiments was expressed in bacteria and lacks glycosylation. The use of IFN-γ expressed in mammalian HEK293 cells (mIFN-γ) and $mgC_{S147-V531}$ also increased IFN-γ-induced ISG expression, suggesting that the effect was not dependent on glycosylation (Supplementary Fig. 7).

We quantified ICAM1 mRNA and protein at different time points post-incubation of HaCaT cells with IFN-γ or $gC_{S147-V531}$ alone or with both. $ICAM1$ mRNA expression level peaked at 4 h post-stimulation (Supplementary Fig. 8a). There was no detectable ICAM1 protein in the absence of IFN-γ treatment or in the presence of $gC_{S147-V531}$ alone at any time post-stimulation (Supplementary Fig. 8b, uncropped blots shown in Supplementary Fig. 18). However, addition of IFN-γ increased ICAM1 protein levels from 4 hours post-stimulation and the combination of IFN-γ and $gC_{S147-V531}$ enhanced IFN-γ-induced ICAM1 transcripts significantly from 6 hours and total protein from 10 h post-incubation, respectively, reaching about threefold more ICAM1 protein at 10 h post-stimulation (Supplementary Fig. 8).

We hypothesised that gC–IFN-γ interaction would also increase ICAM1 at the plasma membrane. We incubated HaCaT cells with IFN-γ alone or together with $gC_{S147-V531}$ and determined the level of ICAM1 at the plasma membrane 24 h later by flow cytometry (Supplementary Fig. 9a, representative gating strategies and histograms/dot plots are shown in Supplementary Information). We also investigated the effect of gC on another ISG, major histocompatibility complex II (MHCII), a known component of the immunological synapse, and determined which gC region was required to modulate IFN-γ-mediated ICAM1 and MHCII levels (Fig. 3). $gC_{S147-V531}$ and $gC_{Y322-S523}$, enhanced IFN-γ-induced levels of ICAM1, while only $gC_{S147-V531}$ had a similar impact on MHCII (Fig. 3a, b). Interestingly, $gC_{Y419-S523}$ did not enhance IFN-γ-mediated induction of ICAM1 and MHCII (Fig. 3a, b). More importantly, $gC_{S147-V531}$ plus IFN-γ increased ICAM1 levels at the plasma membrane of primary normal human epithelial keratinocytes (NHEK), while $gC_{Y419-S523}$ did not (Fig. 3c).

We then determined whether the observed effect was cell-type specific. The addition of IFN-γ and $gC_{S147-V531}$ increased the level of ICAM1 compared to the IFN-γ-only treatment in HaCaT, MeWo, A549 and Jurkat cells, suggesting that the effect was not cell-type dependent (Supplementary Fig. 9b). IFN-γ also increased MHCII levels in HaCaT and MeWo cells, although to a lower extent than ICAM1 (Fig. 3b and Supplementary Fig. 9c). There was no increase in the level of MHCII in A549 and Jurkat cells in any tested conditions, in line with reports from the literature[29,30].

These results showed that VZV gC enhanced IFN-γ-mediated ICAM1 and MHCII protein levels at the plasma membrane in different cell types. The effect on MHCII was considerably lower than that observed for ICAM1. The $gC_{Y419-S523}$ construct did not modify ICAM1 or

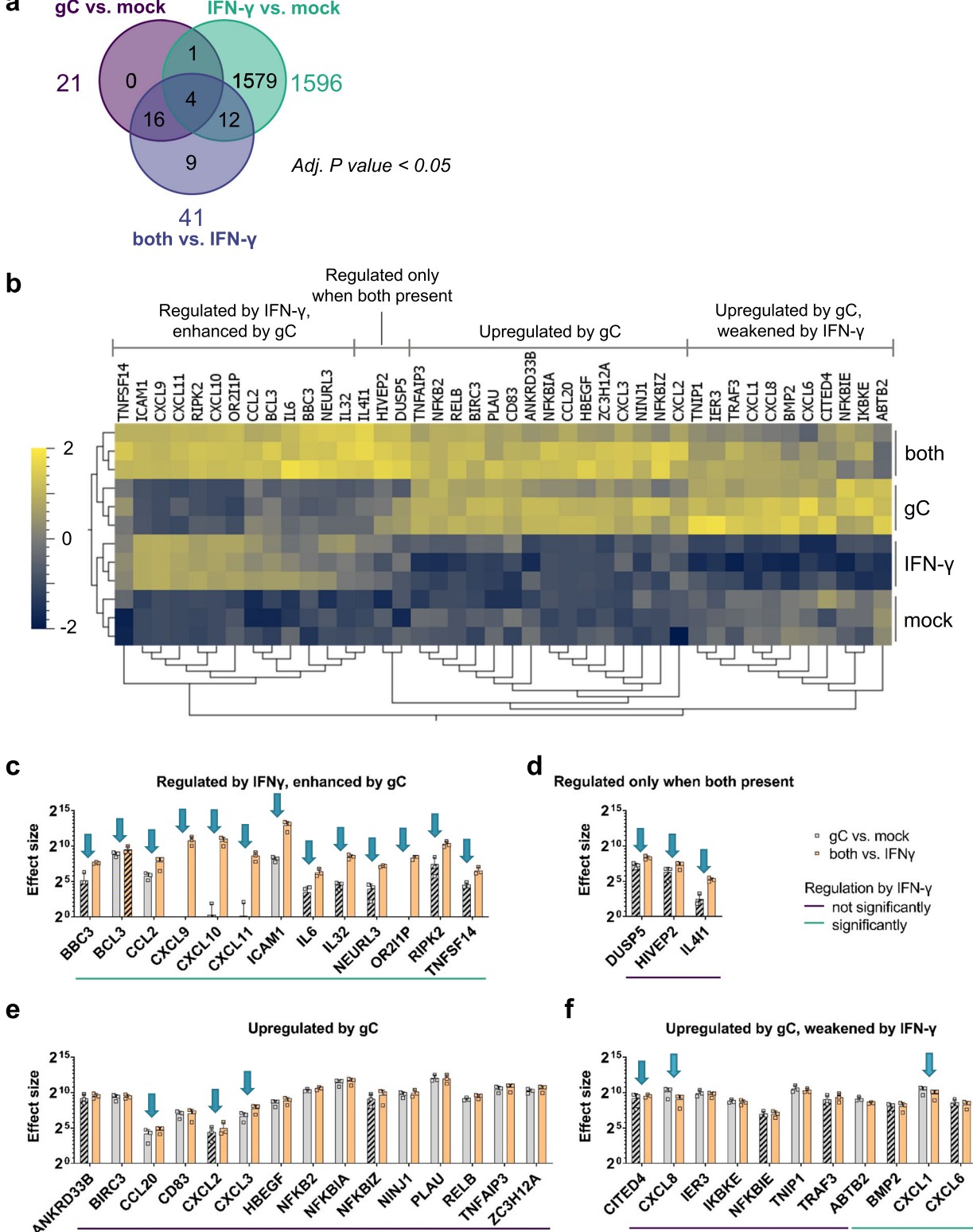

MHCII levels in the presence of IFN-γ. Importantly, the addition of gC$_{S147-V531}$ or gC$_{Y322-S523}$ alone did not significantly enhance the protein levels of ICAM1 or MHCII, suggesting that the effect was not due to the presence of a contaminant.

Taken together with the results presented so far, these results show that VZV gC increases both mRNA and protein levels of ICAM1 in the presence of IFN-γ and confirms an enhancing effect of gC on IFN-γ-

induced ICAM1 expression. In addition, residues Y322-S523 are required for this function.

**gC activity requires signalling through the IFNGR**

The previous results suggested that the mechanism of gC activity involves binding to IFN-γ and signalling through the IFNGR. To confirm this, we employed an antibody that neutralises IFNGR1[31]. The addition

**Fig. 2 | VZV gC induces biased IFN-γ-induced gene expression.** HaCaT cells were stimulated with IFN-γ, gC$_{S147-V531}$, both or mock treated for 4 h. RNA was isolated and further processed for RNAseq. **a** Venn diagram showing the number of genes, whose expression was modified in a statistically significant manner (adjusted $P$ value <0.05) for the three depicted comparisons. Differential gene expression analysis was performed comparing the different treatment conditions. DESeq2 was employed to analyse the RNAseq data and calculate the $P$ values that are used in (**a**–**f**). **b** Normalised counts of genes with an adjusted (adj.) $P$ value <0.05 for either the comparison 'gC vs. mock' or 'both vs. IFN-γ' were plotted as heatmap after calculating the log2 and normalising (mean = 0, variance = 1) using Qlucore Omics Explorer 3.8. Hierarchical clustering was applied to sort for genes with similar behaviour among the treatment conditions. Genes were classified in four different groups based on their expression change upon stimulation with IFN-γ, gC or both.

**c**–**f** Genes with an adj. $P$ value <0.05 for either the comparison 'gC vs. mock' or 'both vs. IFN-γ' were sorted into the four groups identified in the heatmap and the effect sizes were calculated and plotted. Arrows indicate genes that show more than a 1.5-fold change in their effect sizes between both comparisons. The coloured lines below the graphs indicate which genes were significantly regulated by IFN-γ alone. Striped bars indicate that the respective effect size was calculated from a not statistically significant regulated gene in that specific comparison. Panel (**c**) shows the genes regulated by IFN-γ and enhanced by gC. Panel (**d**) shows genes that were regulated when both were present. Panel (**e**) depicts genes mainly upregulated by gC alone and panel (**f**) includes genes that were upregulated by gC and weakened by IFN-γ. Bar charts in **c**–**f** show the mean ± SD of the effect size of $n$ = 3 biological replicates. Data in **c**–**f** are provided as Source Data.

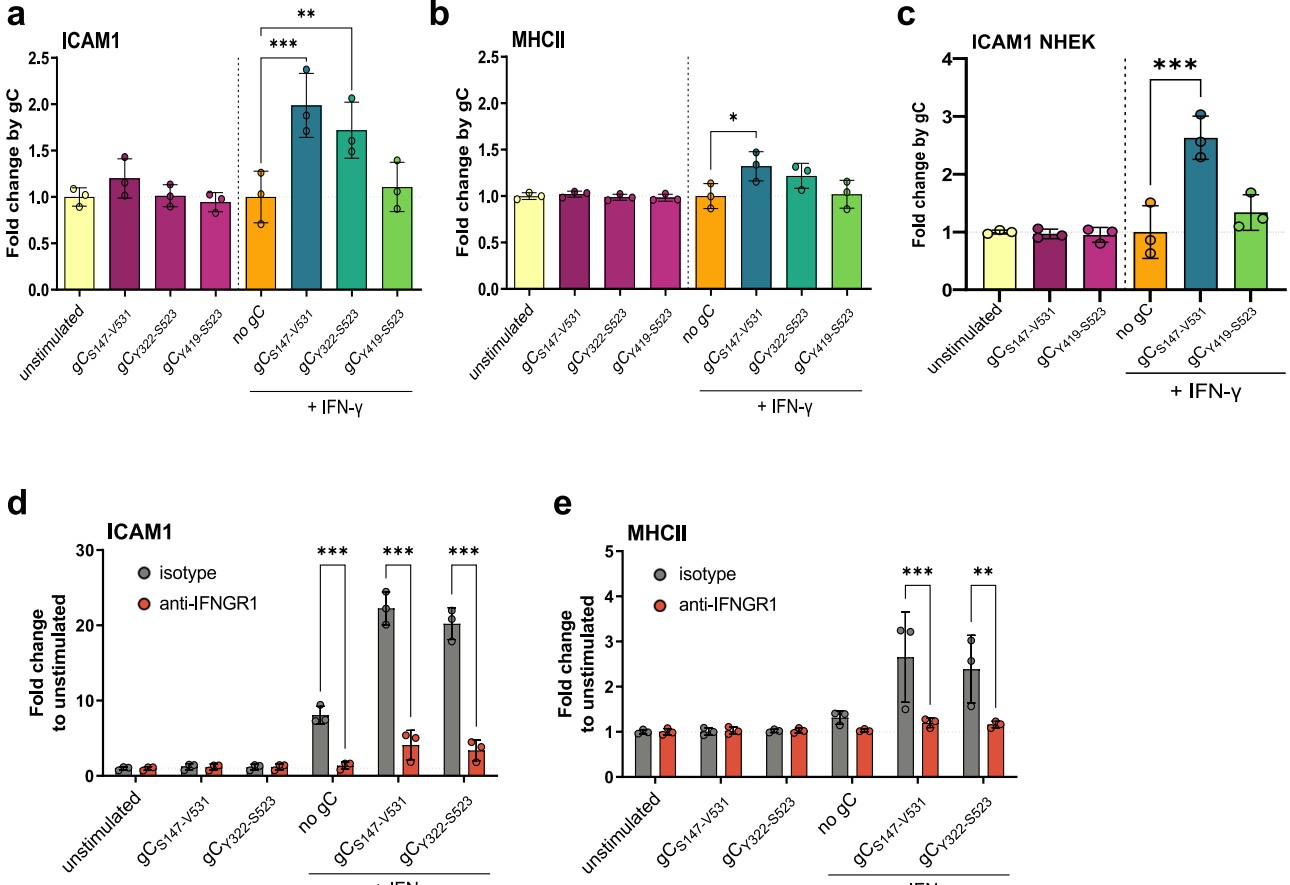

**Fig. 3 | gC enhances IFN-γ-induced ICAM1 and MHCII protein levels at the plasma membrane via IFNGR.** **a**–**c** HaCaT (**a**, **b**) cells or NHEK (**c**) were mock-stimulated or stimulated with 5 ng/mL IFN-γ, 300 nM VZV gC constructs or both for 24 h and then labelled with antibodies binding ICAM1, MHCII, and stained with Zombie-NIR dye. Cells were analysed by flow cytometry and median fluorescence intensities were determined after gating on single alive cells. Bar charts show the fold change of mean ± SD of ICAM1 (**a**, **c**) or MHCII (**b**) surface protein levels induced by gC constructs to either unstimulated or IFN-γ baseline. Each symbol in the graphs shown in (a-c) corresponds to one independent biological experiment ($n$ = 3 biological replicates). **d**, **e** HaCaT cells were pre-treated with 2 µg/mL IFNGR1-neutralising antibody or isotype control for 2 h followed by the addition of 5 ng/mL IFN-γ, 300 nM gC or both for 24 h prior to flow cytometry analysis, as described. Bar charts show the fold change mean ± SD of ICAM1 (**d**) or MHCII (**e**) levels compared to unstimulated cells. Each symbol in the graphs shown in (**d**, **e**) corresponds to one independent biological experiment ($n$ = 3 biological replicates). One-way ANOVA, followed by Šídák's multiple comparisons was performed (comparing condition with gC to baseline without gC (**a**–**c**) and comparing between isotype and neutralising antibody (**d**, **e**)). *$P$ < 0.033; **$P$ < 0.002; ***$P$ < 0.001. Data were provided as Source Data.

of IFN-γ increased ICAM1 and MHCII levels on the plasma membrane of HaCaT cells and the combination of IFN-γ plus gC$_{S147-V531}$ or gC$_{Y322-S523}$ further increased these levels (Fig. 3d, e). The neutralising antibody inhibited IFN-γ activity, and the increase mediated by gC$_{S147-V531}$ or gC$_{Y322-S523}$, while the isotype control did not (Fig. 3d, e).

Since the observed gC activity depended on IFNGR signalling, we addressed the effect of gC on the phosphorylation of signal transducer

and activator of transcription (pSTAT1) at tyrosine (Y) 701. Incubation of HaCaT cells with IFN-γ induced detectable pSTAT1 after 30 minutes, while gC alone did not. The combination of gC$_{S147-V531}$ plus IFN-γ increased STAT1 phosphorylation at both 10 and 30 min (Fig. 4a; uncropped blots shown in Source Data). Similarly, the combination of gC$_{S147-V531}$ and IFN-γ increased pSTAT1 nuclear translocation compared to the cytokine alone (Fig. 4b and Supplementary Figs. 10, 11).

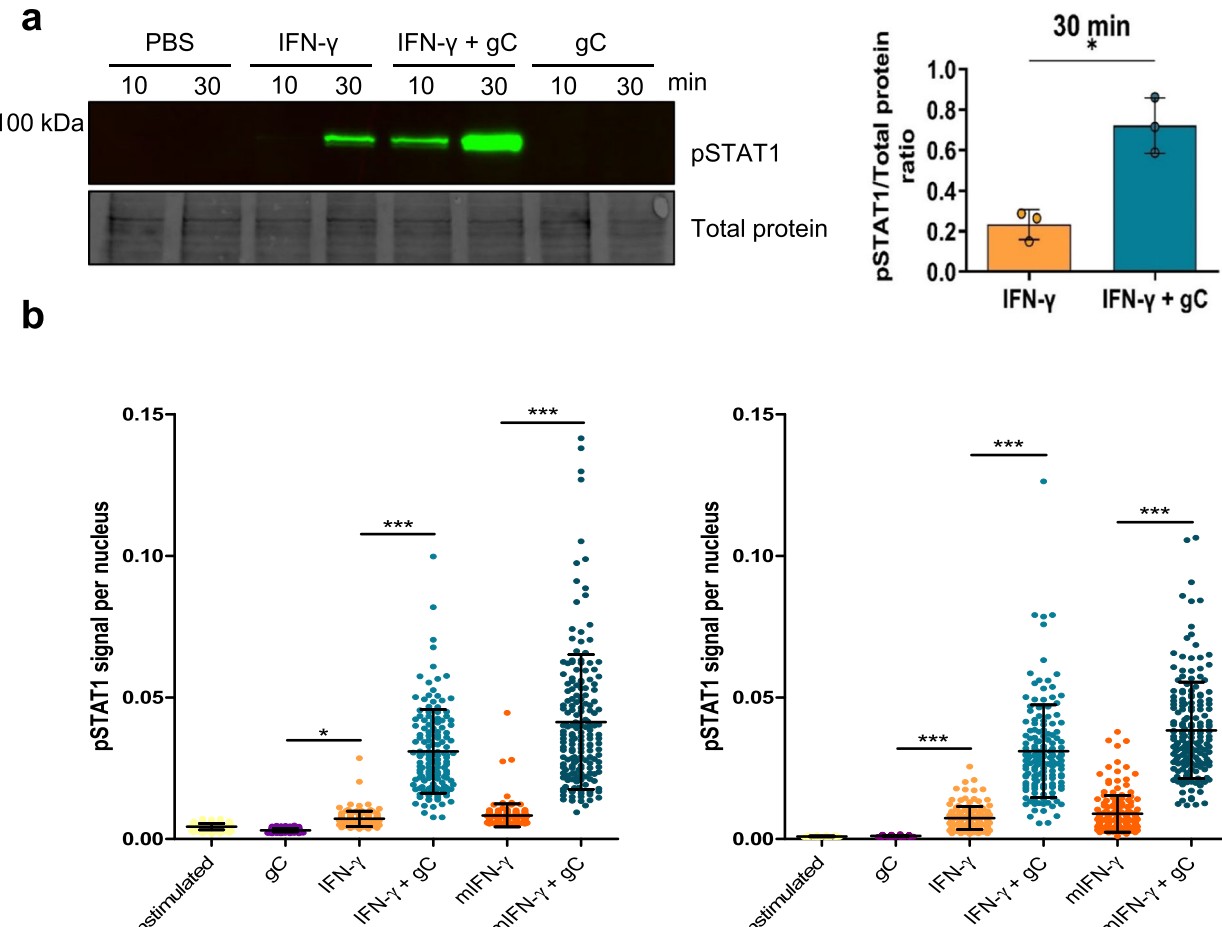

**Fig. 4 | VZV gC enhances IFN-γ-induced phosphorylation of STAT1 and its nuclear translocation. a** HaCaT cells were stimulated with 5 ng/mL IFN-γ, 300 nM VZV gC$_{S147-V531}$ or both and phosphorylation levels of STAT1 (pSTAT1) at Y701 were detected. A representative immunoblot out of three independent ones with its respective loading control (TCE) is depicted, as well as a graph showing the pSTAT1/TCE signal ratios for the 30 min time points (*n* = 3 biological replicates). The three uncropped blots from the biological replicates are shown in Source Data. The graph shows the mean ± SD and the asterisk indicates statistical significance following a two-sided unpaired *t*-test with Welch's correction. **b** HaCaT cells were treated with either 5 ng/mL of IFN-γ produced in bacteria or mammalian cells (mIFN-γ) in the presence or absence of 300 nM gC$_{S147-V531}$. pSTAT1 was detected by immunofluorescence. Representative graphs show nuclear pSTAT1 signal for the two different time points tested (10 or 30 min) from one out of two independent biological replicates (*n* = 2 biological replicates), except for mIFN-γ, which was performed only once. Each dot represents a nucleus and at least 140 nuclei per condition were analysed through Cell Profiler. Error bars indicate mean ± SD and asterisk indicates statistical significance following one-way ANOVA with Bonferroni's post hoc test (*$P < 0.05$; ***$P < 0.001$). Data were provided as Source Data.

We obtained similar results with IFN-γ expressed in mammalian cells (mIFN-γ).

To complement these results, we employed iPSC-derived macrophages from a healthy donor and a patient suffering Mendelian susceptibility to mycobacterial disease due to a deficiency in IFNGR2[32]. VZV productively infects macrophages[33,34], inducing IL-6 expression through Toll-like receptor 2 signalling[35], potentially playing a role in the inflammatory response to VZV and in pathogenesis. iPSC-derived macrophages obtained from a healthy individual expressed higher ICAM1 levels than HaCaT cells and gC increased the effect of IFN-γ in both cell types (Supplementary Fig. 12a). The enhancement of ICAM1 by co-stimulation with gC and IFN-γ occurred with faster kinetics than in the tested cell lines, peaking at 8 h post-stimulation, whereas MHCII induction by IFN-γ was completely abolished by addition of gC (Supplementary Fig. 12b). Lack of IFNGR2 chain abolished the enhancement of ICAM1 surface levels by IFN-γ (Supplementary Fig. 12c). Interestingly, in this cell type, gC alone induced a significant upregulation of ICAM1 at 8 h post-stimulation, independent of signalling via the IFNGR2, by an unknown mechanism.

Overall, these results showed that the synergism of VZV gC and IFN-γ on ICAM1 and MHCII expression required signalling through the IFNGR complex. They also suggest that gC could bind and signal through another receptor to induce ICAM1 protein expression in macrophages, independently of IFNGR signalling.

### VZV gC binds through the glycosaminoglycan-binding site of IFN-γ

The observation that gC did not inhibit IFN-γ and that it required signalling through the IFNGR to increase ICAM1 and MHCII protein levels suggest that gC did not interact with the IFNGR-binding site of IFN-γ. Since IFN-γ also binds to glycosaminoglycans (GAGs), we addressed whether gC interacted with IFN-γ through the GAG-binding regions of IFN-γ by competing the interaction with increasing concentrations of heparin, heparan sulfate and chondroitin sulfate A and B (Supplementary Fig. 13). To do so, we employed gC$_{S147-V531}$, a protein that does not bind GAGs[18]. The presence of soluble GAGs interfered with the ability of gC to bind IFN-γ. The most effective competitor was heparin (Supplementary Fig. 13a), inhibiting 50% of binding with a ratio of 1:0.1 (IFN-γ:GAG; weight:weight ratio), while the other three GAGs required a ratio of about 1:10 (Supplementary Fig. 13b–d). These results suggested that gC bound IFN-γ through the GAG-binding regions of this cytokine.

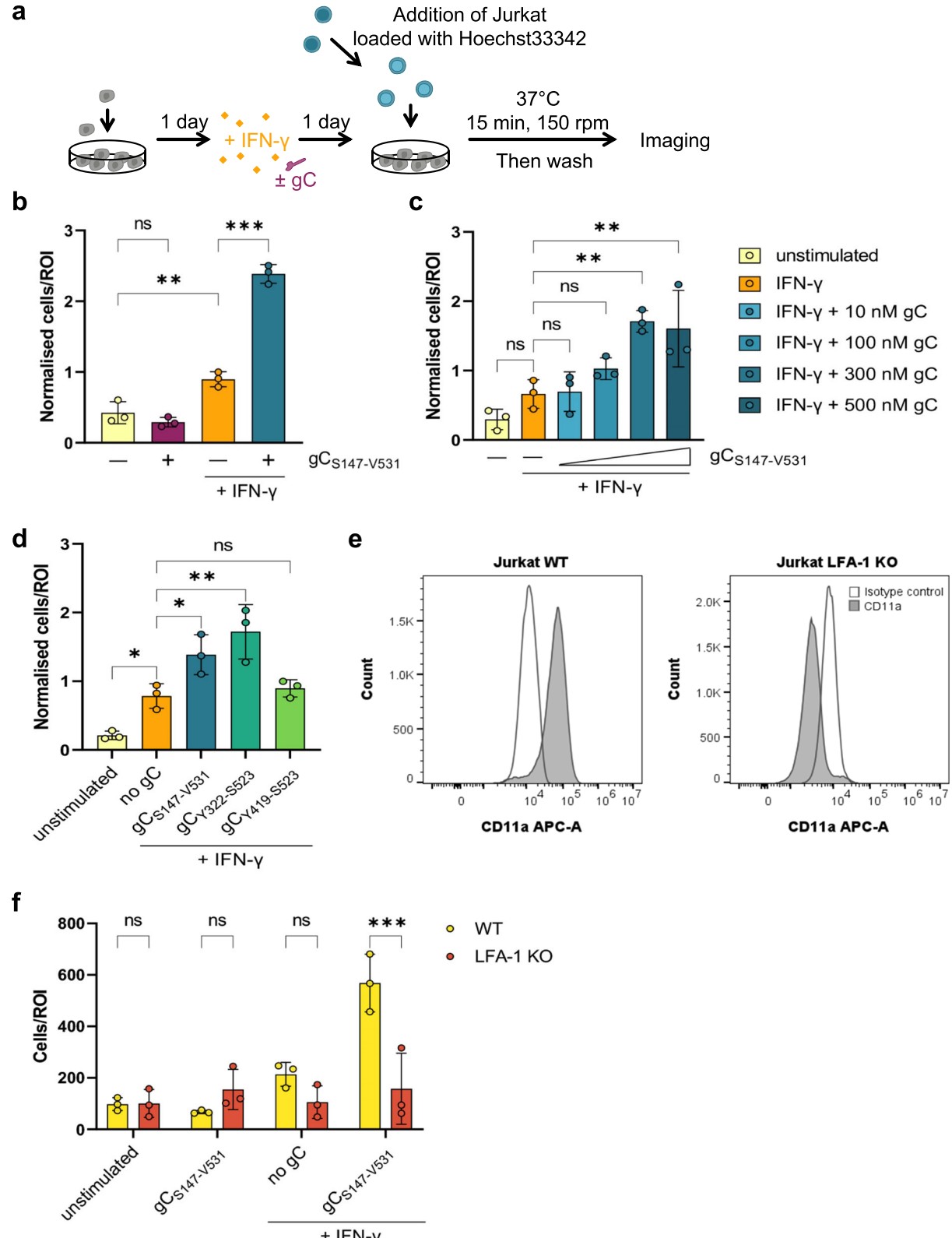

## VZV gC increases IFN-γ-mediated T cell adhesion

ICAM1 binding to LFA-1 on T cells facilitates their adhesion on endo-thelial and epithelial cells. We hypothesised that the increased plasma membrane level of ICAM1 on HaCaT cells upon incubation with IFN-γ and gC could increase T cell adhesion. Therefore, we performed adhesion assays with HaCaT and Jurkat cells (Fig. 5a). Addition of IFN-γ to HaCaT cells increased the number of adhered Jurkat cells by 2.3-fold

compared to the unstimulated control (Fig. 5b). Addition of IFN-γ plus gC_{S147-V531} increased Jurkat cell adhesion by 2.7-fold compared to IFN-γ and 6.1-fold compared to the unstimulated control, while gC_{S147-V531} alone did not. gC_{S147-V531} increased IFN-γ-dependent cell adhesion in a dose-dependent manner when a constant concentration of IFN-γ (5 ng/mL) was employed (Fig. 5c). Both gC_{S147-V531} and gC_{Y322-S523} increased the adhesion of Jurkat to HaCaT cells upon incubation with IFN-γ, while

**Fig. 5 | Co-stimulation of HaCaT cells with IFN-γ and gC increases adhesion of Jurkat cells. a** Schematic representation of the assay. HaCaT cells were seeded 1 day prior to mock-stimulation or stimulation with IFN-γ, gC or both. 24 h after stimulation, Hoechst-labelled Jurkat cells were added and allowed to adhere for 15 min at 37 °C with shaking at 150 rpm. Then, non-adhered cells were washed off and two randomly selected regions of interest (ROI) were imaged per well (triplicate per condition) using an automated microscope (Cytation3, BioTek). The nuclei from adhered cells were quantified using a CellProfiler pipeline. **b**–**d** Adhered Jurkat cells per ROI plotted in a bar chart after normalisation to the overall mean of adhered cells from each assay. Depicted are the comparisons between the four treatment conditions (**b**), the titration of the gC concentration (**c**), and the comparison of the different gC constructs (**d**). If not stated otherwise, 5 ng/mL IFN-γ and 300 nM gC were used. Shown is the mean ± SD. Filled circles represent the values from each independent assay ($n = 3$ biological replicates in **b**–**d**). Ordinary one-way ANOVA followed by Šídák's multiple comparisons test (**b**, to test for preselected pairs) or followed by Dunnett's multiple comparisons test (**c** and **d**, to test each against a control = IFN-γ) were performed. **e** Histograms showing expression of lymphocyte function-associated antigen 1 (LFA-1, CD11a) in Jurkat WT (left) and LFA-1 knockout (KO) cells. **f** Adhesion assay comparing wild-type (WT) Jurkat cells to LFA-1 KO Jurkat cells. Adhered cells per ROI are plotted in a bar chart. Shown is the mean ± SD. Filled circles represent the mean values from each independent assay ($n = 3$ biological replicates, performed in triplicates with 2 ROI per well). Two-way ANOVA followed by Šídák's multiple comparisons test (to test between the two cell types) was performed. ns not significant; *$P < 0.033$; **$P < 0.002$; ***$P < 0.001$. Data in **b**–**f** were provided as Source Data.

gC$_{Y419-S523}$ did not (Fig. 5d), correlating with their binding properties and impact on IFN-γ-mediated ICAM1 expression (Supplementary Fig. 2, 6 and Fig. 3a,c). To determine the role of the ICAM1–LFA-1 interaction in this process, we employed Jurkat cells lacking LFA-1 expression (LFA-1 KO). These cells were previously generated and characterised[36]. We confirmed the lack of LFA-1 protein in the KO cells (Fig. 5e). The absence of LFA-1 resulted in very low adhesion to HaCaT cells in all conditions, irrespective of the presence of IFN-γ and gC$_{S147-V531}$ (Fig. 5f).

Overall, these results indicated that the enhanced IFN-γ-dependent ICAM1 expression induced by gC resulted in higher adhesion of T cells through LFA-1. In line with the ICAM1 upregulation data, these experiments also showed that amino acids Y322-S523 of gC are required for this activity. Importantly, gC did not increase T cell adhesion in the absence of IFN-γ.

## The increase in ICAM1 expression and T cell adhesion requires a stable gC–IFN-γ interaction

The gC$_{S147-V531}$ and gC$_{Y322-S523}$ constructs enhanced ICAM1 expression and T cell adhesion, while gC$_{Y419-S523}$ did not. Moreover, the binding analyses suggest that gC$_{Y419-S523}$ bound IFN-γ worse than gC$_{S147-V531}$ and gC$_{Y322-S523}$ (Supplementary Fig. 2). To better characterise the interaction between the three gC constructs and IFN-γ, we performed multicycle kinetic experiments by SPR and GCI (Supplementary Fig. 14a and Supplementary Tables 1, 2). The curvature of the sensorgrams from kinetic experiments suggested that the interaction between gC and IFN-γ deviates from a simple 1:1 binding, especially for gC$_{S147-V531}$ and gC$_{Y322-S523}$ (Supplementary Fig. 14a). The heterogenous interaction can be described by at least two components, one transient and another more stable (Supplementary Fig. 14b). The transient component contributes more to the interaction at higher than at lower IFN-γ concentrations (Supplementary Fig. 14b) and seems sensitive to increasing salt concentrations (Supplementary Fig. 14c), indicating the relevance of electrostatic interactions. gC$_{Y419-S523}$ interacts with IFN-γ transiently, since the more stable interaction is not observed (Supplementary Fig. 14a, right panel and 14b). The RAPID experiments (Supplementary Fig. 2b) also suggested a weaker binding of gC$_{Y419-S523}$ to IFN-γ, as observed by the very low responses.

Since we observed two modes of interaction with gC$_{S147-V531}$ and gC$_{Y322-S523}$, we employed a heterogenous ligand fit model, while we applied a 1:1 fitting for gC$_{Y419-S523}$ - IFN-γ, since only the transient interaction was detected (Supplementary Fig. 14d and Supplementary Tables 1, 2). The results indicated that both gC$_{S147-V531}$ and gC$_{Y322-S523}$ bound IFN-γ in a similar manner. We observed a transient interaction with an off-rate (Kd1) in the range of $4 \times 10^{-2}$/s and a more stable interaction with a slower off-rate (Kd2) (Supplementary Tables 1, 2). The transient interaction of gC$_{Y419-S523}$ and IFN-γ had a fast off-rate of about $2 \times 10^{-2}$/s, resembling the off-rate of the transient interaction of the longer gC constructs. Together with our functional observations, these data indicate that the stable, higher affinity interaction with IFN-

γ, that occurs only with gC$_{S147-V531}$ and gC$_{Y322-S523}$, is important for the gC function.

## IFN-γ-mediated ICAM1 expression increases in epithelial cells infected with VZV

We obtained all the previous data with recombinant, purified gC constructs. In the next step, we addressed whether gC expressed during VZV infection played a similar role as the purified protein. We initially determined the level of ICAM1 during infection of HaCaT cells using a recombinant bacterial artificial chromosome (BAC)-derived VZV pOka strain expressing monomeric enhanced green fluorescent protein (GFP) under the control of the ORF57 promoter (pOka-Δ57-GFP). The introduction of GFP also deleted the stop codon of ORF58. Both ORF57 and ORF58 are not essential for VZV growth in vitro[37–39]. After two days of infection with pOka-Δ57-GFP, we stimulated the cells with IFN-γ for another day and then quantified ICAM1 protein levels by flow cytometry (Supplementary Fig. 15a). Initially, we gated on live cells for each well, without considering the GFP expression and hence the infection status (Supplementary Fig. 15b). Cells without IFN-γ had low level of ICAM1 expression, irrespective of the presence or absence of VZV in the culture. Upon stimulation with IFN-γ, there was more ICAM1 in all samples, and a tendency toward higher ICAM1 levels in the VZV- than in mock-infected cultures (Supplementary Fig. 15b). In the VZV-inoculated cultures, we discriminated between uninfected bystander cells and productively VZV-infected GFP positive (GFP$^+$) cells. We separated the infected cells into GFP$^{high}$ and GFP$^{low}$ populations, resembling cells with high and low viral replication and viral gene expression, respectively. IFN-γ treatment slightly reduced the number of infected cells, decreasing the percentage of GFP$^{high}$ cells and slightly increasing that of GFP$^{low}$ cells compared to the mock-treated control, due to the inhibitory effect of IFN-γ on VZV replication, as previously shown[40,41] (Supplementary Fig. 15c). Comparing the fold change by IFN-γ, we observed a significant higher fold change of ICAM1 in the VZV-inoculated culture, independent of the cells being uninfected bystanders or productively infected cells (Supplementary Fig. 15d). The productively infected cells showed higher fold change in ICAM1 upon addition of IFN-γ than the uninfected bystander cells, and this was more pronounced in the GFP$^{low}$ cells (Supplementary Fig. 15e). Interestingly, ICAM1 induction was lower in GFP$^{high}$ cells, indicating high virus production, in line with previous results showing that VZV infection inhibits ICAM1 expression[42,43]. Taken together, these results indicate that VZV-infected HaCaT cells express more ICAM1 than mock-infected cells upon IFN-γ stimulation.

## VZV gC increases T cell adhesion during infection

To investigate the relevance of the IFN-γ-mediated ICAM1 upregulation in the context of infection and to address the role of gC, we employed two recombinant, BAC-derived VZV pOka viruses: pOka-gC-GFP and pOka-ΔgC-GFP. In both viruses, the monomeric enhanced GFP signal is expressed from the promoter of *ORF14* (encoding gC) (Supplementary

Fig. 16). Since *ORF14* is a late gene, only expressed after VZV DNA replication, the presence of GFP indicates productive infection. VZV pOka-ΔgC-GFP was previously generated and characterised[18]. To generate pOka-gC-GFP, we fused GFP to *ORF14* using *en passant* mutagenesis[44], similar to the generation of the pOka-ΔgC-GFP (Supplementary Fig. 16a). We addressed whether the insertion of GFP could modify the location of gC in infected epithelial ARPE-19 cells. VZV gC and gC-GFP localised predominantly at the plasma membrane of cells infected with parental pOka and pOka-gC-GFP (Supplementary Fig. 16b). Both GFP-expressing viruses replicated with similar kinetics in HaCaT cells (Supplementary Fig. 16c), indicating that lack of gC did not affect VZV replication in vitro, as previously shown[18,45]. Moreover, the level of GFP detected was similar for both viruses. Therefore, we used GFP detection as a surrogate of productive VZV infection.

We then performed cell adhesion assays on HaCaT cells infected with the same plaque-forming units (PFU) of pOka-gC-GFP or pOka-ΔgC-GFP. At two days post-infection (dpi), we stimulated the cells with IFN-γ for another day and added Hoechst-labelled Jurkat cells. We imaged the cells and measured the mean GFP and Hoechst intensities (Fig. 6a). Analysis of the GFP levels at 3 dpi showed that both viruses replicated similarly in HaCaT cells (Fig. 6b), also after the addition of IFN-γ. This indicated once again that gC did not reduce the antiviral effect of IFN-γ. Since the number of counted nuclei correlated with the mean Hoechst signal, we used the mean fluorescence intensities as a surrogate for adhered cells. The number of infected cells may vary between region of interests (ROI). Therefore, we normalised the Hoechst signal to the respective GFP signal for each ROI. With increasing IFN-γ concentrations, we observed a slight increase in cell adhesion upon infection with both viruses with a tendency of higher Jurkat adhesion in the cells infected with pOka-gC-GFP compared to those infected with pOka-ΔgC-GFP (Fig. 6c). We then calculated the ratio of the Hoechst/GFP value from pOka-gC-GFP to the pOka-ΔgC-GFP and normalised it to mock-treated cells (Fig. 6d). This analysis showed that there was an IFN-γ dose-dependent increase in Jurkat cell adhesion when HaCaT cells were infected with VZV pOka-gC-GFP compared to pOka-ΔgC-GFP. Notably, the adhered cells clustered around the productively infected cells (GFP+ cells; Fig. 6e), but also adhered to the bystander cells. This observation is in line with the ICAM1 upregulation assay, where the GFP^low and the bystander cells expressed higher ICAM1 levels than the GFP^high cells (Supplementary Fig. 15).

Overall, these results suggest that the expression of gC during VZV infection of HaCaT cells facilitates the adhesion of Jurkat cells.

### VZV gC facilitates the spread from epithelial cells to T cells

We hypothesised that the higher ICAM1 expression and T cell adhesion mediated by the combination of IFN-γ and gC may result in more efficient VZV spread to T cells. We performed an adhesion experiment in which the co-culture of HaCaT and Jurkat cells was maintained for 1 to 2 days after cell adhesion. Then, the GFP levels in both populations were analysed by flow cytometry. The amount of GFP+ HaCaT cells was similar between both gC-expressing and ΔgC viruses, with ~21–23% of infected cells in untreated cultures (Fig. 7a). The amount of GFP+ HaCaT cells for both viruses declined to about 17 and 12%, upon incubation with 0.5 and 5 ng/mL of IFN-γ, respectively. In the untreated co-culture, the amount of GFP+ Jurkat cells, indicative of VZV infection, was approximately 2.5% for the pOka-gC-GFP virus and about 2.0% for the pOka-ΔgC-GFP virus (Fig. 7b). We observed an IFN-γ-dependent decrease of GFP+ Jurkat cells for both viruses, but a tendency indicating less infected Jurkat cells for the pOka-ΔgC-GFP compared to the pOka-gC-GFP. Since only infected HaCaT cells can transmit the virus and the amount of infected HaCaT cells decreased with IFN-γ treatment, we calculated the ratio of GFP+ Jurkat cells to GFP+ HaCaT cells (Fig. 7c). Again, we observed a tendency of better transmission for the pOka-gC-GFP virus compared to the pOka-ΔgC-GFP virus. To better judge the

effect of IFN-γ treatment, the ratio between pOka-gC-GFP to pOka-ΔgC-GFP was calculated from the Jurkat/HaCaT ratio to obtain the fold change by gC (Fig. 7d). These data indicated that the pOka-gC-GFP virus spreads better than the pOka-ΔgC-GFP from HaCaT to Jurkat cells in the presence of IFN-γ.

To increase the relevance of our results, we repeated these experiments with human peripheral blood mononuclear cells (PBMCs) obtained from two healthy donors. As before, the percentage of GFP+ HaCaT cells dropped for both viruses upon addition of IFN-γ in a dose-dependent manner (Fig. 7e). However, while pOka-gC-GFP virus spread from HaCaT cells to PBMCs, the pOka-ΔgC-GFP virus barely spread at all to PBMCs (Fig. 7f). This was even more obvious, when calculating the GFP+ PBMC/HaCaT ratio and the fold change by gC, showing that the pOka-gC-GFP virus spread better to PBMCs than the pOka-ΔgC-GFP virus (Fig. 7g,h).

Taken together, the results strongly suggested that gC is critical to facilitate virus transmission from HaCaT to PBMCs, in both the absence and presence of exogenous IFN-γ.

## Discussion

Viruses have evolved a variety of mechanisms to inhibit or modulate the innate and adaptive immune responses to establish productive infection and spread in the host, particularly under low virus input. One such strategy, employed by herpes- and poxviruses, consists of the expression of type I transmembrane or secreted viral proteins that bind cytokines to modulate or alter their activities. On most occasions, such viral proteins act as decoy receptors, inhibiting the immune response[46,47]. However, here we show that the vCKBP gC binds IFN-γ, without apparently inhibiting IFN activity and the stimulation of ISGs. On the contrary, gC binding to IFN-γ led to the induction of an IFN-γ-mediated biased gene transcription program in which the expression of a select group of ISGs was more upregulated than under IFN-γ alone. This set included *ICAM1*, which was increased at the mRNA and protein levels in keratinocytes, facilitating Jurkat cell adhesion and resulting in better infection of both Jurkat cells and PBMCs. We hypothesise that the gC–IFN-γ interaction facilitates VZV spread from epithelial to T cells, both at the respiratory lymphoid epithelium and prior to secondary viremia, when mononuclear cells are recruited to the infected skin[5–7].

VZV gC combined with IFN-γ induced differential expression of a subset of genes. Some of these genes, including chemokines *CCL2*, *CCL20*, *CXCL2*, *CXCL3*, *CXCL9*, *CXCL10* and *CXCL11*, as well as the adhesion molecule *ICAM1*, play key roles in chemotaxis, an activity that VZV gC also enhances[18]. VZV gC also increased IFN-γ-mediated induction of other genes like *IL4I1*, whose protein product downregulates the immune response[28]. Secretion of IL4I1 at the immunological synapse decreases TCR activation and signalling, inhibiting T cell proliferation and facilitating the development of regulatory T cells[28]. The increased ICAM1 levels and its interaction with LFA-1 could also affect T cell differentiation. Signalling through ICAM1 tends to result in the expression of pro-inflammatory cytokines[48], and LFA-1 activation in the context of TCR signalling is relevant for T cell differentiation into effector phenotypes[49]. We do not yet know whether the differential gene expression due to gC-IFN-γ interaction modulates T cell differentiation and/or activity. This could be relevant in the context of VZV infection of T cells and the subsequent VZV-directed modulation of their receptor expression profile that results in T cells with a skin-homing, effector memory phenotype[4]. Interestingly, despite the infected T cells having an effector phenotype[4], they do not seem to eliminate the virus, but rather mediate the transport of VZV to the skin and facilitate efficient spread into keratinocytes[5]. We speculate that the gC IFN-γ binding activity could also impact gene expression in T cells at later stages of VZV pathogenesis, such as the release of virus at the skin and during infection of T cells from keratinocytes prior to secondary viremia[5–7].

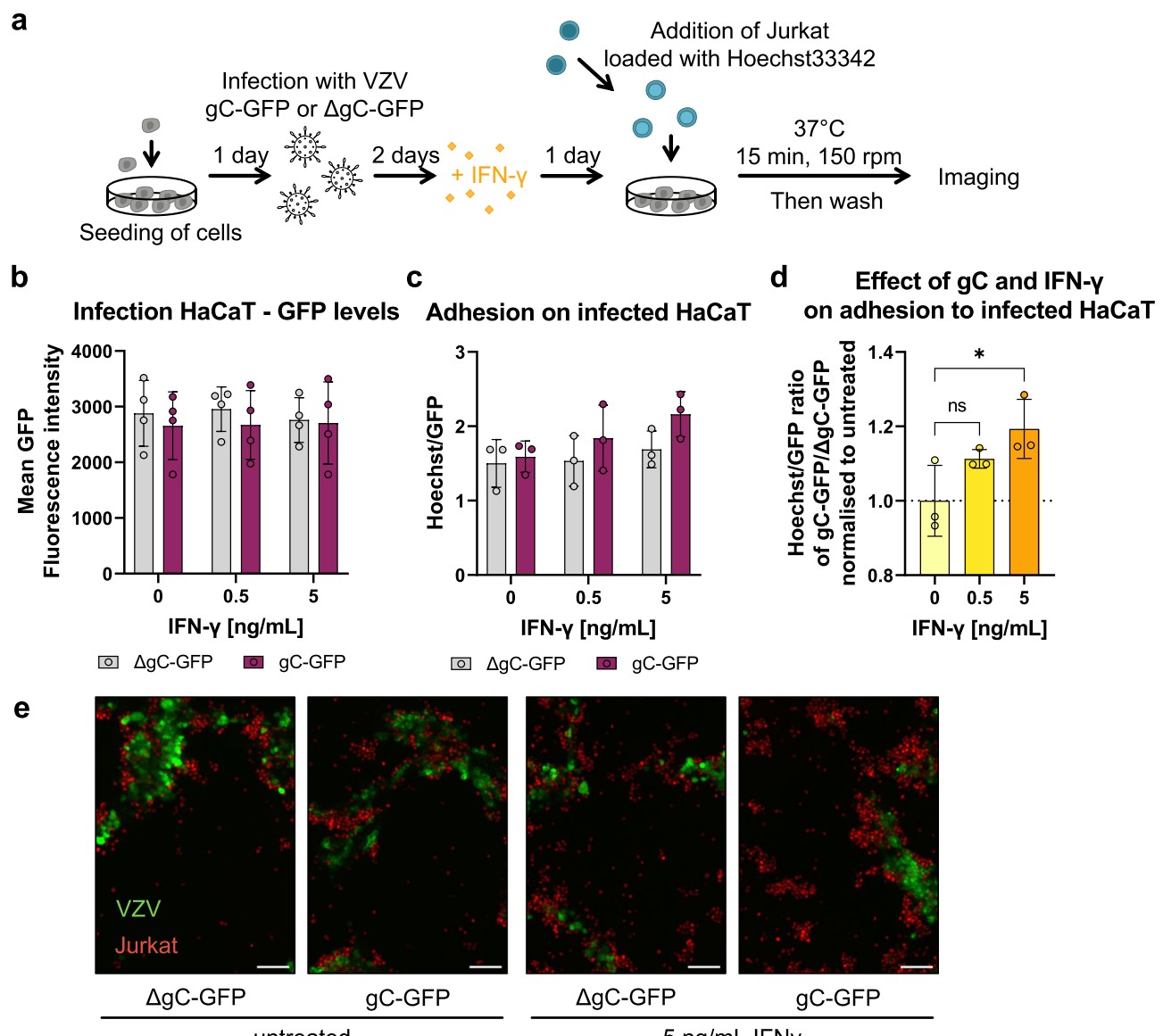

**Fig. 6 | Jurkat cells adhere better to cells infected with the VZV-gC-GFP virus than with VZV-ΔgC-GFP. a** Schematic representation of the assay. HaCaT cells were seeded 24 h prior to infection with VZV pOka expressing GFP fused to gC (gC-GFP) or expressing GFP instead of gC (ΔgC-GFP). 48 h after infection, the cells were stimulated with IFN-γ or mock-treated. The next day, Hoechst-labelled Jurkat cells were added and allowed to adhere for 15 min at 37 °C on a shaking platform at 150 rpm. Then, non-adhered cells were washed off and two randomly selected regions of interest (ROI) were imaged per well (triplicate per condition) using an automated microscope (Cytation3, BioTek) and the mean Hoechst and GFP intensities were determined. **b–d** Each circle corresponds to one independent experiment (*n* = 4 biological experiments in (**b**) and 3 in (**c** and **d**)). Shown are the mean ± SD of the independent experiments. **b** Graph showing mean GFP fluorescence intensity obtained from HaCaT cells infected with VZV-gC-GFP or VZV-ΔgC-GFP and incubated or not with IFN-γ. **c** Bar chart showing the amount of adhered Jurkat cells normalised to the amount of infected HaCaT cells (Hoechst/GFP ratio) in the presence or absence of IFN-γ. **d** Graph showing the Hoechst/GFP ratio from HaCaT cells infected with VZV-gC-GFP divided by that of HaCaT cells infected with VZV-ΔgC-GFP and normalised to the mock-treated condition. Ordinary one-way ANOVA followed by Dunnett's multiple comparisons test (to test each against a control = no IFN-γ). ns not significant; *$P < 0.033$; **$P < 0.002$; ***$P < 0.001$. Data in **b–d** are provided as Source Data. **e** Representative fluorescence microscopy images of Jurkat and HaCaT cells in the four experimental conditions. The GFP signal corresponding to productive infection is depicted in green, whereas the adhered Hoechst positive cells are shown in red. The scale bar corresponds to 100 μm.

We validated the enhancement of ICAM1 at the mRNA and protein levels, the latter also in NHEK cells, and showed that ICAM1 interaction with LFA-1 was required for the increased T cell adhesion observed in the presence of gC and IFN-γ compared to IFN-γ alone. The glycosylation pattern of both gC and IFN-γ did not affect gC activity. Infection with VZV expressing gC also led to enhancement of ICAM1 levels and T cell adhesion. This enhancement resulted in a trend of higher VZV spread from HaCaT to Jurkat cells and, especially, to PBMCs. In future studies, we will determine whether gC also improves spread from NHEK to PBMCs and which leucocyte subtype is prominently infected.

It has been previously shown that VZV-infected T cells express higher levels of LFA-1, and it was speculated to facilitate migration[4]. The increased level of LFA-1 could also facilitate VZV spread to keratinocytes through an immunological synapse-like formation. Some viruses, including human immunodeficiency virus and HSV-1, transform the immunological synapse into a virological synapse to facilitate cell-to-cell spread[50,51].

IFN-γ inhibits VZV replication in different cell types[41], even more efficiently than IFN-α[40]. In vivo, IFN-γ and other cytokines secreted by VZV-specific CD4 T cells reduce VZV pathogenesis during primary

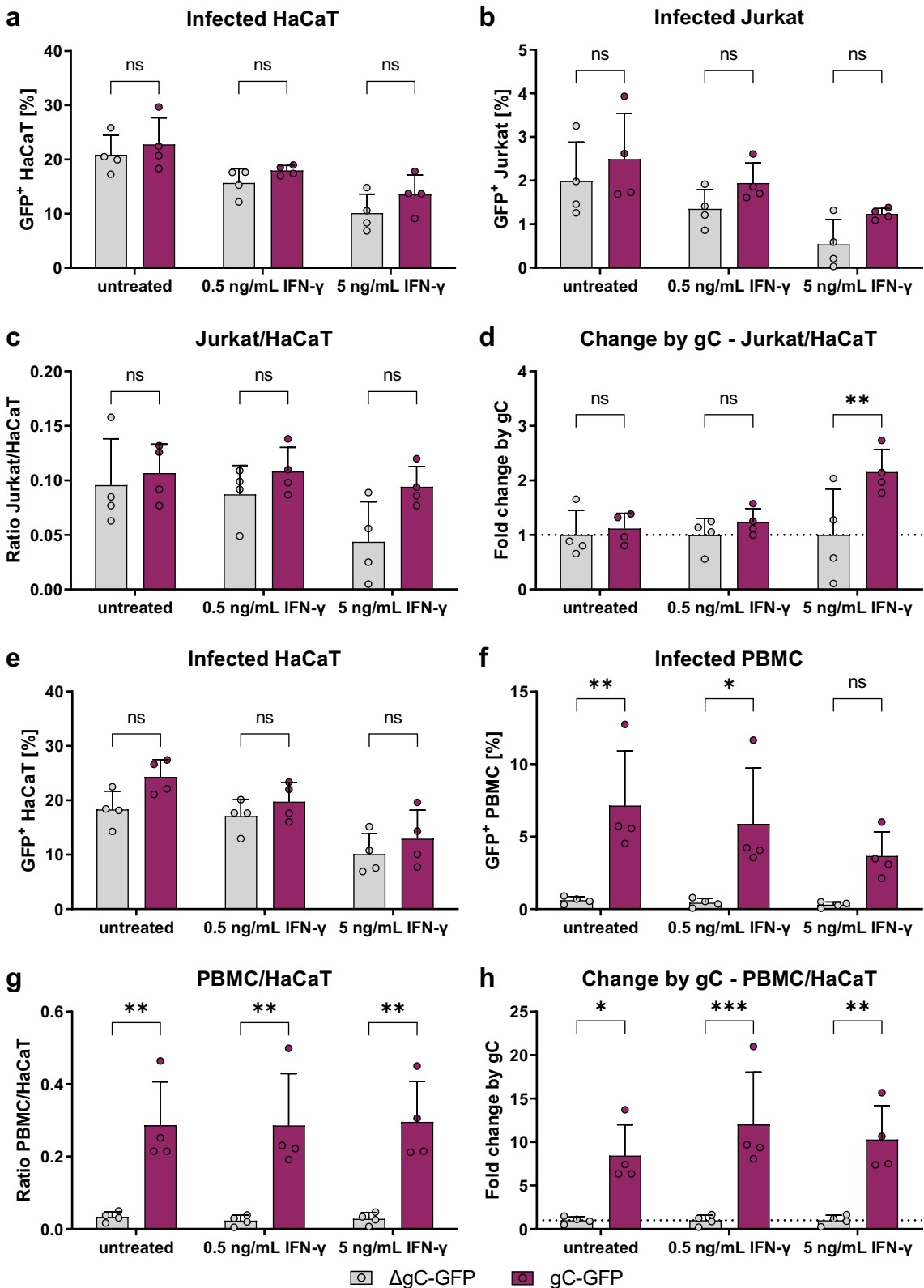

**Fig. 7 | VZV-gC-GFP spreads more efficiently from HaCaT cells to Jurkat cells and PBMCs than VZV-ΔgC-GFP.** Quantification of infected HaCaT, Jurkat cells and PBMCs by flow cytometry. Each symbol in the graphs (**a**–**h**) corresponds to one independent biological experiment (*n* = 4 biological replicates) depicted as mean ± SD. Magenta indicates cultures infected with VZV-gC-GFP and grey indicates VZV-ΔgC-GFP infected cultures. **a, b** Percentages of GFP⁺ HaCaT cells (**a**) and GFP⁺ Jurkat cells (**b**) after co-culture are plotted as bar charts. **c** Ratio of GFP⁺ Jurkat cells to GFP⁺ HaCaT cells is plotted as bar chart. **d** The fold change attributed to the

presence of gC is plotted as a bar chart. **e, f** Percentages of GFP⁺ HaCaT cells (**e**) and GFP⁺ PBMCs (**f**) after co-culture are plotted as bar charts. **g** Ratio of GFP⁺ PBMCs to GFP HaCaT cells is plotted as bar chart. **h** The fold change attributed to the presence of gC is plotted as a bar chart. Two-way ANOVA followed by Šídák's multiple comparisons test (to test between the two viruses) was performed in all experiments. ns not significant; *$P$ < 0.033; **$P$ < 0.002; ***$P$ < 0.001. Data were provided as Source Data.

infection and upon reactivation[52,53]. Several VZV genes inhibit IFN-γ function[54–57]. Interestingly, VZV gC did not seem to reduce the antiviral activity of IFN-γ, which inhibited VZV replication in HaCaT cells irrespective of the presence of gC. Similarly, IFN-γ reduced the spread of the virus from HaCaT to Jurkat cells and PBMCs. However, the virus expressing gC spread better to Jurkat cells and PBMCs than the ΔgC virus, even in the presence of exogenous IFN-γ. Therefore, the presence of gC during infection conferred an advantage to spread from HaCaT to both Jurkat cells and PBMCs. VZV gC also facilitates VZV spread between human keratinocytes in skin xenografted in SCID mice[21]. During infection of human skin, gC is highly expressed[58] and gC expression is higher in highly virulent viruses than in an attenuated vaccine strain[45,59]. All these results highlight the role of gC as a virulence factor. We have previously shown that gC is present in the viral particle[18] and our current results indicate that it also locates at the plasma membrane. Therefore, gC could potentially interact with IFN-γ on the viral particle and on the surface of infected cells, followed by binding to IFNGR on the surface of epithelial cells, including those already infected. Since VZV gC is expressed with late kinetics, its function would occur late during infection, at a time when virions are already being produced.

An interesting and unsolved question is how VZV gC induces biased signalling through IFNGR. Most viral IFN-binding proteins bind to the cytokine through its receptor-interacting site in order to block IFN activity. Our results suggest that VZV gC binds to IFN-γ through its GAG-binding domain, and this would allow interaction of the cytokine with its cognate receptor. Alternatively, it is possible that gC may also bind and signal through another receptor that coordinates with IFNGR to mediate an altered signalling. As a surface glycoprotein that is located on the virion surface and on the plasma membrane, it is likely that gC binds directly to receptors on the cell surface, including those that influence IFN-γ activity. Our data with neutralising antibodies to IFNGR show that signalling through this receptor is required for gC enhancement of ICAM1 expression in epithelial cells. Moreover, although gC alone enhanced the expression of a few genes, the effect size was not large enough to result in observable phenotypic changes in ICAM1 protein levels and cell adhesion. gC induced phosphorylation of STAT1 at Y701 and its nuclear translocation only in the presence of IFN-γ and not when added alone. Finally, gC$_{Y419-S523}$ bound IFN-γ only transiently and did not increase its activity, suggesting that stable interaction with the cytokine through gC residues Y322-S523 is required to enhance ICAM1 expression in several cell lines and, thereby, T cell adhesion. Overall, these results suggest that gC activity requires high-affinity interaction with IFN-γ and signalling through the IFNGR. However, gC increased ICAM1 protein independently of IFN-γ in human iPSC-derived macrophages, suggesting that gC can also act independently of IFN-γ in these cells. Moreover, VZV gC increased spread from HaCaT cells to PBMCs even without the exogenous addition of IFN-γ. This could also point to the existence of an unknown interaction of gC with a receptor expressed in these cells. Alternatively, it could also be due to the presence of IFN-γ secreted by PBMCs. VZV gC could form oligomers that bind several IFN-γ dimers, increasing its net valency. However, this would not explain why only the expression of a reduced number of ISGs is increased. Identification of potential transcription binding sites in the promoters of the 42 significantly upregulated genes did not reveal a clear consensus for specific transcription factors. Based on the already known biased agonists of IFNGR[15], we hypothesise that gC induces conformational changes on IFN-γ that lead to different interactions with IFNGR chains, followed by modifications of signalling pathways and differential gene expression. To address these hypotheses, we are currently investigating gC mechanism of action and seeking to obtain the crystal structure of gC bound to IFN-γ.

Overall, our results advance the knowledge in the field of viral immunomodulation by the discovery of a viral protein that binds and modulates IFN-γ activity, acting as the only yet known viral IFN-γ biased agonist, increasing the expression of few ISGs, including ICAM1 and leading to higher T cell adhesion and VZV spread to lymphocytes. This is highly relevant due to the importance of lymphocyte infection for VZV spread and pathogenesis. Moreover, IFN-γ-based therapies are complicated by the large number of ISGs induced and their pleiotropic activities. The information obtained here could serve to design specific IFN-γ agonists that have therapeutic potential, as previously suggested[15].

## Methods

### Ethics statement
Peripheral blood mononuclear cells (PBMCs) from anonymised healthy blood donors were provided by the Institute of Transfusion Medicine, Hannover Medical School, Germany. The donors signed a consent for the use of small amounts of their blood for research purposes, approved by the Ethics Committee of Hannover Medical School #2519-2014 and #10476_BO_K_2022. Foreskin normal human epidermal keratinocytes (NHEK) provided by the Department of Dermatology and Allergy, Hannover Medical School, Germany, were obtained from surgical residuals from anonymous children. No ethical permit was required. Generation of human iPSC was approved by the local Ethics Committee of Hannover Medical School (2127-2014).

### Cytokines
Recombinant cytokines (IFN-α, IFN-β, IFN-γ, IFN-λ1, IFN-λ2, IFN-ω and TNF-α) were purchased from PeproTech, with the exception of IFN-γ expressed in HEK293 cells (mIFN-γ), that was obtained from R&D Systems (Minnesota, USA). The different IFN-α subtypes (IFN-α1, IFN-α2, IFN-α4, IFN-α5, IFN-α6, IFN-α7, IFN-α8, IFN-α10, IFN-α14, IFN-α16, IFN-α17 and IFN-α21) have been previously described[60] and were kindly provided by Ulf Dittmer (University Hospital Essen).

### Construct design
The coding sequences of different gC constructs were introduced into a pMT vector backbone[61] for expression in Drosophila S2 cells. The genes were inserted downstream a Drosophila BiP signal sequence to direct proteins to the endoplasmic reticulum and all constructs had a double strep-tag for efficient protein purification. All gC constructs were amplified from the VZV Dumas strain (Accession Number GenBank: X04370.1) using primers shown in Supplementary Table 3. The gC$_{P23-V531}$ and gC$_{S147-V531}$ were cloned using a restriction enzyme-based strategy (BglII and SpeI sites) as described before[18]. The gC$_{P23-V531}$ resembles the His-tagged rSgC previously reported[18] while the gC$_{S147-V531}$ lacks the 7 N-terminal residues present in the previously reported IgD-Strep[18]. To generate gC$_{Y322-S523}$ and gC$_{Y419-S523}$ we performed restriction-free cloning[62] using primers shown in Supplementary Table 3. To express gC$_{S147-V531}$ in HEK293ExPi, the corresponding coding sequence was cloned into pcDNA3.1 (Thermo Fisher Scientific, MA, USA).

### Production of recombinant proteins
For expression and purification of the soluble gC constructs, stable S2 transfectants were established and proteins produced as described previously[63], with minor modifications. Briefly, 2 μg of gC expression plasmid was co-transfected with 0.1 μg pCoPuro plasmid[64]. Following a 6-day selection period with puromycin (8 μg/mL), stable cell lines were adapted and grown in insect-Xpress media (Lonza). For large-scale production, supernatant from cells incubated with 4 μM CdCl$_2$ for 5 days was collected and soluble protein was purified by affinity chromatography using a Strep-Tactin XT 4Flow column (IBA Lifesciences) followed by size exclusion chromatography using a HiLoad 26/600 superdex 200 pg column (Cytiva) equilibrated in 20 mM HEPES pH 7.4 and 150 mM NaCl at 2 mL/min flow rate. For cell culture assays, purified proteins were buffer exchanged to PBS using a

superdex 200 increase 10/300 column (Cytiva) equilibrated with PBS at 0.5 mL/min flow rate. Protein purity was analyzed on Coomassie-stained SDS-PAGE gels or using stain-free 2,2,2-trichloroethanol (TCE) gels[65]. Fractions containing pure protein were concentrated, flash-frozen, and stored at -80 °C until required for further analysis.

To express $gC_{S147-V531}$ in HEK293ExPi cells, the cells were transfected with pcDNA3.1- $gC_{S147-V531}$ using ExpiFectamine 293 transfection reagent (Thermo Fisher Scientific) as per the manufacturer's recommendation albeit with minor method modifications. Briefly, a plasmid-DNA cocktail containing the $gC_{S147-V531}$-expressing plasmid, and plasmids encoding cell cycle inhibitors p21 and p27, as well as the large T antigen of the SV40 virus at ratios of 0.69:0.05:0.25:0.01, respectively, was prepared in 5 ml Opti-MEM (Gibco) at a concentration of 1 µg/ml of final culture volume. The ExpiFectamine 293 transfection reagent was diluted in 5 ml Opti-MEM and allowed to incubate for 5 min at room temperature prior to mixing with the plasmid-DNA cocktail. Following a 20-min incubation at room temperature, the transfection mixture was added dropwise to the cells followed by a 16-h incubation before addition of enhancers. Transfected cells were then incubated for another 4 days, after which cells were pelleted and the soluble $gC_{S147-V531}$ was affinity purified from the supernatant as described for proteins from insect cells.

### Biophysical protein–protein interaction studies
**Surface plasmon resonance (SPR).** The Biacore X100 and S200 systems were used to investigate protein–protein interactions by SPR. All experiments were performed with 1× HBS-EP as a running buffer, if not stated otherwise, and at 25 °C. A pH-scouting was performed to find optimal conditions of protein immobilisation. To do so, the protein was diluted at least 1:10 in sodium acetate buffer with different pH (4.5, 5 and 5.5) or maleate pH 5.9 and injected onto the chip surface, which was afterwards washed using 50 mM NaOH. Immobilisation was performed using the amine coupling kit and wizard according to the manufacturer's instructions, leading to the covalent coupling of protein ligands on CM4 or CM5 chips. Depending on the purpose the target level was specified. Flow cell 1 (FC1) was always used as a negative control. For screening experiments and GAG-competition assays, the association time was set to 90 s and the dissociation time was set to 60 s, and 100 nM analytes were injected at a flow rate of 10 µL/min. For GAG-competition assays the analyte was mixed with increasing amounts of GAG (w:w ratio). For kinetics experiments, the association time was set to 90 s and the dissociation time was set to 420 s, and analytes were injected at a flow rate of 30 µL/min and in a concentration ranging from 0.39 to 50 nM. After each analyte injection, the surface was regenerated using 10 mM glycine pH 2. In each run, two to three cycles with 1× HBS-EP were included as a control and for subtraction of the blank signal. Analysis and sensorgram adjustment were performed using the Biacore X100 or S200 Evaluation software. The FCx−FC1 was calculated, and a blank injection was subtracted to obtain the specific signal for the binding to the ligand.

**Grating-coupled interferometry (GCI).** The Creoptix WAVEdelta system was used to investigate protein–protein interactions by GCI. DXH chips were used to analyse the gC–IFN interaction. These chips have a dextran surface similar to the CM4/5 chips from the Biacore system and can be used for covalent immobilisation by amine coupling. In the first step, the chip was conditioned with borate pH 9 to ensure proper hydration of the surface. A pH-scouting was performed to find optimal conditions for capture and immobilisation. Both, pH-scouting and immobilisation were performed following a similar protocol as for the Biacore system but using 0.2× HBS-EP (filtered and degassed) as running buffer and a specific immobilisation time, set according to the desired immobilisation level and based on the response obtained during pH-scouting. For kinetic experiments, the RAPID wizard for tight binders (fr 50 µL/min, acq 1 Hz, ass 120 s, diss 1800 s) was used. As a running buffer for binding and kinetic experiments filtered and degassed 1× HBS-EP was used.

### RNA isolation
To isolate RNA, the NucleoSpin® RNA kit (Macherey and Nagel) was used for general-purpose and RNAseq experiments in which cells were cultured in a six-well format. To isolate RNA from cells cultured in 96-well plates, the Quick-RNA™ MicroPrep kit from Zymo was used. Both kits were used according to the manufacturer's instructions. The RNA concentration was quantified photometrically using the NanoDrop 1000. RNA was stored at −80 °C or, if possible, directly processed to obtain cDNA as explained in the "RT-qPCR" section, below.

### RT-qPCR
RT-qPCR was performed following a two-step protocol. For cDNA synthesis, the LunaScript® RT Super Mix Kit from NEB was used according to the manufacturer's instructions. For qPCR analysis, the Luna® Universal qPCR Master Mix from NEB was used according to the manufacturer's instructions with a total reaction volume of 10 µL. For gene expression analysis, 1 ng RNA/cDNA was used as a template and amplified with oligonucleotides shown in Supplementary Table 4. The qTower from Analytic Jena with the qPCRsoft 4.1 software was used. The ΔΔCt-method was used to analyse the data using the qPCRsoft 4.1 (Analytik Jena) Software following the Livak method. Actin was used for normalisation.

### RNA-Seq
HaCaT cells were seeded at a density of $0.8 \times 10^6$ cells/well of a six-well plate to obtain a confluent cell layer the next day. For stimulation, a 10× stimulation mix of IFN-γ and/or $gC_{S147-V531}$ was prepared in 1× DPBS and preincubated for 30 min at 37 °C. The medium was replaced by 1350 µL growth medium and 150 µL 10× stimulant was added for 4 h. The final concentrations of IFN-γ and $gC_{S147-V531}$ were 5 ng/mL and 300 nM, respectively. Incubation of the cells was performed in a humidified incubator at 37 °C and 5% $CO_2$. Afterwards, the RNA was isolated as described above. RNA quality control was performed by the RCU Genomics using a 2100 Bioanalyzer (Agilent).

Total RNA was extracted at 4 hours post-treatment to reduce the potential impact of feedback loops in the results and subsequently enriched for polyadenylated RNA. Following sequencing, quality control, alignment of reads to the host genome, and the generation of gene abundance counts, library generation, quality control, and sequencing, as well as BCL to FASTQ conversion, were performed by Novogene (Cambridge, UK). Briefly, mRNA was purified from total RNA using poly-T oligo-attached magnetic beads. After fragmentation, the first strand cDNA was synthesised using random hexamer primers, followed by the second strand cDNA synthesis using dTTP for a non-directional library (NEB Next® Ultra RNA Library Prep Kit for Illumina®). After end repair, A-tailing, adaptor ligation, size selection, amplification, and purification, the fragment length distribution of individual libraries was monitored using an Agilent 2100 Bioanalyzer. Quantification of libraries was performed by use of Qubit and real-time PCR.

Equimolar amounts of individually barcoded libraries were pooled for a paired-end 150 bp sequencing run on an Illumina NovaSeq6000 S4 (300 cycles). The clustering of the index-coded samples was performed according to the manufacturer's instructions. Raw reads of FASTQ format were first processed by Novogene and reads containing adaptors, poly-N, and low-quality nucleotides were removed.

Raw data processing was conducted by use of nfcore/rnaseq (version 1.4.2; National Genomics Infrastructure at SciLifeLab Stockholm, Sweden), using the bioinformatics workflow tool Nextflow to

pre-processed raw data from FastQ inputs, align the reads and perform quality-control. The genome reference and annotation data were taken from GENCODE.org (*Homo sapiens*: GRCh38.p13; release 34).

Quality control and processing of sequencing raw data (bulk RNAseq experiment) was conducted by the Research Core Unit Genomics (RCUG) at Hannover Medical School.

Normalisation and differential expression analysis were performed on the internal Galaxy (version 20.05) with DESeq2 (Galaxy Tool Version 2.11.40.6)[66] with default settings except for 'Output normalised counts table', which was set to 'Yes' and all additional filters were disabled ('Turn off outliers replacement', 'Turn off outliers filtering', and 'Turn off independent filtering' set 'Yes'). Analysis was performed with multiple levels of primary factor (all different stimulations conditions) comparing all levels against each other.

R version R-4.1.2 (managed through RStudio version 2021.09.2 + 382) and Excel were used to sort for interesting genes based on adjusted *P* values and fold changes. Additionally, Excel was used to calculate the effect sizes from the normalised gene counts and fold changes thereof.

### Protein precipitation

Acetone precipitation was used to recover the proteins from samples used for RNA isolation according to the manufacturer's instructions for the RNA isolation kit (Quick-RNA™ Microprep Kit from Zymo). Therefore, 1 volume of the collected flow-through from RNA isolation was mixed with 4 volumes of pre-chilled acetone (−20 °C) and incubated at −20 °C for at least 30 min. Then, samples were centrifuged at 17,000×*g* for 10 min and the SN discarded. The protein pellet was washed with ethanol and after another spin at 17,000×*g* for 1 min and removal of the SN, the pellet was air-dried for 10 min at RT. Afterwards, the pellet was resuspended in 20 µL 5× SDS sample buffer with β-mercaptoethanol and subjected to SDS-PAGE and western blotting.

### Detection of ICAM1 and MHCII by flow cytometry

Adherent cells were seeded ($1 \times 10^5$ HaCaT or NHEK, $2.5 \times 10^5$ MeWo and $0.625 \times 10^5$ A549 cells) a day in advance into 48-well plates and incubated for 18–24 h at 37 °C, 5% $CO_2$ in humidified incubator. Jurkat cells were passaged 1:2 and the day of the experiment $2 \times 10^5$ cells were seeded into 48-well plates. The iPSC-derived macrophages were seeded at $0.15 \times 10^6$ cells/48-well and cultivated in RPMI1640 supplemented with 10% FCS, 1% Pen/Strep, and 50 ng/mL huM-CSF (Peprotech) for 4 days and then stimulated.

Stimulation mix containing IFN-γ, gC or both, was prepared in 1× DPBS and preincubated for 30 min at 37 °C, then growth medium was added. The medium was removed from the cells and the stimulation medium was added. To neutralise IFNGR signalling, cells were incubated with 2 µg/mL IFNGR1-neutralising antibody (BD Bioscience #557531) or an isotype control (BD Bioscience #554721) for 2 h prior to stimulation. The stimulation mix was added to the antibody-containing medium. Stimulation of cell lines lasted 24 h while that of iPSC-derived macrophages was performed during 2, 5, 8, 12 and 24 h, both at 37 °C, 5% $CO_2$ in a humidified incubator. To analyse cells by flow cytometry, the cells were detached with accutase, and washed twice with 1× DPBS + 1% FCS (and 2.5 mM $Ca^{2+}$ if annexin V staining was included to quantify apoptotic cells). Cells were kept on ice after detachment and stained for 30 min with anti-ICAM1-PE (Biolegend #353106) and anti-MHCII-FITC (ImmunoTools #21629233×2) or isotype control antibodies (Biolegend #401606, eBioscience™ (Invitrogen) #12-4714-82), and Zombie-NIR (Biolegend #423106) to quantify cell viability. Afterwards, cells were washed again twice with DPBS + 1% FCS (and 2.5 mM $Ca^{2+}$ for the first wash if annexin V was included) and resuspended in DPBS + 1% FCS for measurement at the flow cytometer (Cytoflex, Beckman Coulter). Macrophages were rinsed off at the end of the stimulation time and stained for flow cytometric analysis.

To determine ICAM1 levels during infection, $1 \times 10^5$ HaCaT cells were seeded into 12-well plates. The next day, cells were infected with $7.5 \times 10^4$ PFU HaCaT-associated VZV Δ57-GFP reporter virus in a volume of 500 µL infection medium for 2 h at 37 °C and 5% $CO_2$ in a humidified incubator. Afterwards, the inoculum was removed, cells were washed with 1× DPBS, and fresh infection medium was added. Cells were further incubated for 2 days. Then, cells were stimulated with 5 ng/mL IFN-γ or mock-treated for 24 h. Cells were detached using accutase and stained for 30 min with anti-ICAM1-APC (Biolegend #353111) and Zombie-NIR (Biolegend #423106). After the last washing step, cells were resuspended in 100 µL 4% PFA and incubated on ice for 15 min with mixing every 5 min. Then, cells were washed once more with DPBS + 1% FCS and resuspended for analysis at the Cytoflex. For analysis, first, the cell population was included, then doublets and dead cells were excluded. The remaining population was gated for a GFP⁻ = uninfected (or uninfected bystander in VZV-inoculated cultures) population and a GFP⁺ = productively infected population. Based on the GFP levels, we further distinguished between a GFPhigh and GFPlow population within the infected cells.

### Generation of recombinant VZV

Recombinant, BAC-derived pOka-ΔgC-GFP, pOka-gC-GFP and pOka-Δ57-GFP were employed in this study. pOka-ΔgC-GFP was described before[18] and contains an enhanced monomeric GFP instead of *ORF14*. Recombinant VZV expressing gC-GFP protein (pOka-gC-GFP) was generated by en passant mutagenesis through the addition of a 5 × alanine linker followed by monomeric enhanced GFP (GFP) to the 3' end of *ORF14* in the background of previously generated BAC-pOka strain[44,67]. The BAC-pOka was mutated by insertion of 5 x Alanine linker-GFP-Kanamycin resistance (KanR) cassette in which an excisable KanR gene disrupts the GFP ORF. The KanR is flanked by a duplicated fragment of the GFP sequence and I-*Sce*I restriction sites, which allows subsequent excision of KanR and the seamless repair of the GFP ORF by Red recombination in *E. coli* strain GS1783[44]. The cassettes were amplified by PCR with the plasmid pEP-GFP-in[68] as a template and using the following primers to fuse the gC with the GFP gene in the BAC. Fwd 5′-TATCGCAGTTATCGCAACCCTATGCATCCGTTGCTGTT-CAGCAGCAGCA GCAGCAATGGTGAGCAAGGGCGAGGA-3′ and Rev- 5′-AAAATGATATACACAGACGCGTTTGGTTGGTTTCTGTTTACTTGTA-CAGCTCGTCCATG-3′.

The recombinant pOkaDXRR57DG (here called pOka-Δ57-GFP) was developed by BAC mutagenesis using the pOka DXRR BAC recently published[69] that has corrections for two spurious mutations found in the original pOka BAC DX[67]. An mCherry Kan-in cassette with an internal *Isce*I was kindly provided by Dr Gregory Smith (Northwestern University Chicago, IL). Two oligos were used to create a new complete deletion of ORF57 and the insertion of the GFP-kan in gene: 57FF 5′-AAAATACTTTGACCGACCAACCAATTAATACTGAAAAT AGCGGTC**ATG**GACGTACGAGAACGTAAT*GTGAGCAAGGGCGAG*-3′ and 57 R 5′-GATTATATTTAACGGCTTTTAATTTGAAGACACCTATCCTCTG *ACATCACTTGTACAGCTCGTCCAT*-3′. The design of the deletion primers was such that 18 nucleotides of the beginning of the ORF57 gene are retained (ORF57ATG in bold) as a fusion to the fluorescent gene so that the deletion does not affect the small region of ORF58 C-terminus that overlaps ORF57 underlined. The sequence denoted in italics is homologous to the fluorescent gene. The purified PCR amplified fragment was electroporated into bacterial strain GS1783 containing the pOka DXRR after the induction of the recombination enzymes, and recombination was selected by gain of kanamycin resistance as recently detailed by ref. 69 and previous publications from the Kinchington Laboratory.

Positive colonies were then grown, induced for I-*Sce*I expression by the addition of Arabinose, and subjected to a heat induction to induce the λRec Recombination enzymes. Following plating on plates lacking kanamycin but containing Arabinose, colonies were then

screened for loss of kanamycin resistance by replica plating. All BACs were verified for insertion into the correct site by RFLP analysis and sequencing across junctions.

To reconstitute infectious recombinant viruses, MeWo cells were transfected with fresh BAC DNA using Lipofectamine 2000 (Invitrogen) or TransIT-X2 (Mirus) in a six-well plate (10 µl Lipofectamine + up to 100 µl of BAC-VZV DNA). Lipofectamine–DNA complexes were produced in Opti-MEM and added onto subconfluent (~80%) MeWo cells in a dropwise manner. After 24 h, the medium was changed and cells were incubated with maintenance splits of cells every week until the formation of syncytia.

To ensure the genomic integrity and lack of undesired mutations, the viruses were sequenced by Illumina and Oxford Nanopore Technologies following reconstitution in mammalian cells (see Supplementary Methods).

### Cell adhesion and virus spread assays

**Adhesion assay.** About $3 \times 10^4$ HaCaT cells were seeded into a 96-well plate to reach confluency the next day. Then, Jurkat cells were split 1:2 and the HaCaT cells were stimulated with IFN-γ +/− recombinant gC at concentrations indicated in the respective figure legends (if not stated otherwise, 5 ng/mL IFN-γ and 300 nM gC were used). The stimulants were prepared in 1× DPBS, preincubated for 30 min at 37 °C, then growth medium was incorporated and the mixture was added to the cells following removal of the old medium. Triplicates were prepared for each condition.

For adhesion on infected cells, $1.5 \times 10^4$ HaCaT cells were seeded per well of 96-well plates to obtain a subconfluent cell layer. The next day cells were incubated with $7.5 \times 10^2$ PFU of BAC-derived VZV pOka-gC-GFP or ΔgC-GFP in infection medium for 2 h at 37 °C. Then, the inoculum was pipetted up and down and removed, followed by a washing step with 1× DPBS. Next, the infection medium was added and the cells were incubated in a humidified incubator at 37 °C and 5% $CO_2$. After 2 days, cells were stimulated with different concentrations of 0, 0.5 or 5 ng/mL IFN-γ (prepared as 10× in 1× DPBS and diluted in infection medium). Triplicates were prepared for each condition.

After 24 h of stimulation, Jurkat cells were loaded with 5 µg/mL Hoechst 33342 and resuspended in RPMI supplemented with bivalent cations (1 mM $Ca^{2+}$ and 2 mM $Mg^{2+}$ final concentration) to $1.5 \times 10^5/85$ µL. The HaCaT cells were washed two times with 1× DPBS. Then, 85 µL/well of labelled Jurkat cells was added, and the plate was immediately incubated for 15 min at 150 rpm and 37 °C. Afterwards, the non-adhered cells were taken off, and the remaining cells were washed four times with DPBS supplemented with bivalent cations (1 mM $Ca^{2+}$ and 2 mM $Mg^{2+}$ final concentration). Then, the adhered Jurkat cells were imaged at Cytation3 (BioTek) with two non-overlapping images per well (DAPI channel). Nuclei were counted using the CellProfiler Pipeline.

**Virus spread assay.** About $5 \times 10^5$ HaCaT cells were seeded per well of a 48-well plate to obtain a subconfluent cell layer. Cells were inoculated 24 h later with $2 \times 10^3$ PFU BAC-derived VZV pOka-gC-GFP or ΔgC-GFP in infection medium for 2 h at 37 °C, 5% $CO_2$ in a humidified incubator. Afterwards, the inoculum was pipetted up and down and removed, followed by a washing step with DPBS. Next, the infection medium was added and the cells were incubated in a humidified incubator at 37 °C and 5% $CO_2$. Two days later, the infected HaCaT cells were stimulated with 0, 0.5 or 5 ng/mL IFN-γ (300 µL/well) for 1 day. The day before the start of the co-culture, Jurkat cells were passaged 1:2, and PBMCs were thawed and cultivated for 1 day in RPMI supplemented with 10% heat-inactivated FCS, 1× L-glutamine, 1× sodium pyruvate and 1× Pen/Strep at a density of $2 \times 10^6$ cell/mL at 37 °C and 5% $CO_2$ in a humidified incubator.

Before starting the co-culture, Jurkat cells and PBMCs were labelled with Hoechst 33342. The final cell pellet was resuspended at a density of $5 \times 10^5$ cell/250 µL. The infected HaCaT cells were washed two times with 1× DPBS, 250 µL of labelled Jurkat cells or PBMCs were added, and the plate was immediately incubated for 15 min at 150 rpm and 37 °C. Afterwards, the non-adhered cells were taken off and the remaining cells were washed carefully four times with 500 µL DPBS supplemented with bivalent cations (1 mM $Ca^{2+}$ and 2 mM $Mg^{2+}$ final concentration). Then, a 1:1 mix of DMEM supplemented with 2% heat-inactivated FCS, 1× L-glutamine, and 1× Pen/Strep and RPMI supplemented with 2% FCS, 1× L-glutamine, and 1× Pen/Strep medium was added and the cells were co-cultured at 37 °C and 5% $CO_2$ in a humidified incubator for 24–48 h.

To analyse the spread of VZV to Jurkat and PBMCs, all cells of the co-culture were collected and analysed by flow cytometry using the Cytoflex. Therefore, the medium, wash solution and trypsin/EDTA-detached cells were collected in FACS tubes. Cells were washed twice with FACS buffer (1% heat-inactivated FCS and 2 mM EDTA in 1× DPBS) and then the pellet was resuspended in 300 µL FACS buffer containing 2.7% PFA (1:2 mix FACS buffer:4% PFA) and incubated for at least 30 min on ice. Afterwards, the GFP and Hoechst signals were measured at the Cytoflex.

### Statistical analysis

Unless stated in the specific methods or figure legends, statistical analysis was performed using GraphPad Prism. The dispersion measures, as well as the type of performed tests are indicated in the figure legends. Significances are indicated in the figures as ns not significant; $^*P < 0.033$; $^{**}P < 0.002$; $^{***}P < 0.001$.

### Reporting summary

Further information on research design is available in the Nature Portfolio Reporting Summary linked to this article.

## Data availability

The RNAseq datasets generated and analysed in the current study are available in the European Nucleotide Archive repository, with the following accession number PRJEB61951. The genome sequences of the viruses generated in this report are available at GenBank with the following accession numbers: pOka-Δ57-GFP, PP378487; pOka-ΔgC-GFP, PP378488; pOka-gC-GFP, PP378489. iPSC lines used in this study can be obtained after signing appropriate material transfer agreements. RNA fold changes comparing the different groups and their significances were calculated using DESeq2 can be found in Source Data. All data presented in graphs within figures is included in Source Data. The uncropped western blots for Fig. 4a can be found in Source Data and the uncropped gels and western blots for Supplementary Figs. 1, 8 can be found in Supplementary Information (Supplementary Figs. 17, 18, respectively). The gating strategy and the dot plots are presented as Supplementary Information. Source data are provided with this paper.

## Code availability

We did not create new codes in this study. All codes employed are previously published and we have included the appropriate references.

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

## Acknowledgements

We thank Khanh Le-Trilling and Mirko Trilling (University of Duisburg-Essen, Germany) for their critical discussion and suggestions. We thank Beate Sodeik, Martin Messerle (Institute of Virology, Hannover Medical School, Germany) and Thomas Pietschmann (Twincore, Hannover, Germany) for providing HaCaT, Jurkat E6.1 and A549 cell lines, respectively. We thank Ilona Klug and Katinka Döhner (Department of Dermatology and Allergy, Hannover Medical School, Germany) for the NHEK cells and protocols to culture them. Our gratitude also goes to Carsten Münk (Heinrich-Heine University Düsseldorf, Germany), for providing the Jurkat LFA-1-KO cells. We thank Kathrin Sutter and Ulf Dittmer (University Hospital Essen, Germany) for providing the different IFN-α subtypes and Edward Fitzgerald (Malvern Panalytical, Creoptix), Anja Drescher (Cytiva, Biacore) and Rudolf Bauerfeind (Research Core Unit for Laser Microscopy, Hannover Medical School, Germany) for technical support. We are grateful to Nikolaus Osterrieder (Freie Universität Berlin, Germany, and Cornell University, USA) and Benedikt Kaufer (Freie Universität Berlin, Germany) for the parental BAC-derived VZV pOka strain. We are grateful to Gregory Smith (Northwestern University Chicago, IL) for providing an mCherry Kan-in cassette with an internal *Isce*I site for BAC mutagenesis. We thank Britta Eiz-Vesper (Hannover Medical School, Germany) for providing the blood to isolate PBMCs and all blood donors who agreed to the use of small amounts of blood for research. We thank the Research Core Unit Genomics (RCUG) at Hannover Medical School for performing quality control and processing of sequencing raw data (bulk RNAseq experiment). This work was funded by the Deutsche Forschungsgemeinschaft (DFG, German Research Foundation) under Germany's Excellence Strategy—EXC 2155—project number 390874280 (https://www.resist-cluster.de/en/), by the Deutsche Forschungsgemeinschaft (DFG, German Research Foundation)—SFB 900/3—project number 158989968 to AV-B (TPB9) and TK (TPB10) (https://www.mh-hannover.de/sfb900.html) and by the Deutsche Forschungsgemeinschaft (DFG, German Research Foundation) in the framework of the Research Unit FOR5200 DEEP-DV (443644894) project VI 762/4-1 to A.V.-B. C.J. and G.S. were funded by the Deutsche Forschungsgemeinschaft (DFG, German Research Foundation) project number 405772731 and GS by the German Federal Ministry of Education and Science (BMBF) via the Research network University Medicine (NUM; projects 'COVIM' (FKZ: 01KX2021). N.P. was funded by the Deutsche Forschungsgemeinschaft (DFG, German Research Foundation) project number 500627539. C.J., N.P., G.S., S.B., J.W. and L.M.-M. were supported by the Hannover Biomedical Research School (HBRS) and the Center for Infection Biology (ZIB). L.M-M. was supported by a fellowship of the German Government in the course of the German 'Excellence Initiative' donated by the HBRS and the ZIB. PRK was supported by NIH awards AI158510, AI151290, P30-EY08098, and unrestricted awards from the Research to Prevent Blindness Inc, NY, and the Eye & Ear Foundation of Pittsburgh. Generation of the GMP line LiPSC-GR1.1 was supported by the NIH Common Fund Regenerative Medicine Programme. The NIH Common Fund and the National Center for Advancing Translational Sciences (NCATS) are joint stewards of the LiPSC-GR1.1 resource. This work was supported by the Deutsche Forschungsgemeinschaft (DFG, German Research Foundation) LA 3680/9-1 and 10-1) (N.L.)); the European Research Council (ERC) under the European Union (EU)'s Horizon 2020 research and innovation program (Grant agreement 852178); and the EU (Grant agreement 101100859 and 101158172) (N.L.). The views and opinions expressed are, however, those of the authors only and do not necessarily reflect those of the EU or the ERC. Neither the EU nor the granting authority can be held responsible for them.

## Author contributions

C.J., N.P., G.S., S.B., L.M.-M., J.W., K.A.K., V.G.-M. and A.V.-B. performed the experiments and/or analysed the data. L.S., B.R. and H.B. provided

technical assistance. C.R.-G., E.N., P.R.K. and N.L. provided relevant material and reagents. D.P.D. and C.J. analysed the transcriptomics data. P.R.K., N.L., T.K. and A.V.-B. obtained funding. T.K. and A.V.-B. supervised the work. C.J. and A.V.-B. wrote the original manuscript draft. C.J., N.P., L.M.-M., G.S., P.R.K., D.D., T.K. and A.V.-B. corrected the draft.

## Funding

## Competing interests
The authors declare no competing interests.
