## [Peer Review File · Nature Communications]

Viral modulation of type II interferon increases T cell adhesion and virus spreadREVIEWER COMMENTS

Reviewer #2 (Remarks to the Author):

For Authors

The author team is composed of excellent investigators from diverse disciplines who have come together with an integrated body of work. The virus studied, VZV, is medically significant despite the (limited) availability of good vaccines. The cell types studied, or at least their in vivo counterparts, specifically epithelial cells and lymphocytes, are highly relevant to the pathophysiology of primary VZV infection. The work builds on prior, related work by this team, for example <https://journals.plos.org/plospathogens/article?id=10.1371/journal.ppat.1006346>

High level the paper first identifies a novel viral protein-host protein binding interaction in vitro, second presents in vitro discovery data from bulk RNAseq datasets of the possible impact of this binding interaction, third present one-cell protein level data to correlate with the mRNA data, and finally present two-cell data which aims to back up the functional significance of the first three steps. The overall model is actually quite novel: rather than expressing a simple IFN-sequestering protein as an immune evasin, as some poxviruses do, the authors propose that that a viral protein actually binds and increases the biological activity of IFN γ to in turn influence cells to upregulate an adhesion molecule that is an ISG (ICAM1) and thus favor cell to cell transmission of VZV from one cell to another. If validated, this would be quite an original finding. There is no simple in vivo model for VZV infection, such that in vitro studies may be quite valuable before progressing to the artificial immune deficient murine models or attempting to study the issues addressed in NHP with SVV, a homolog of VZV. However, there are significant limitations with each of the major findings which call into question how well the overall thesis of the work is supported by the data.

The first data presented concerns gC binding to IFN γ . The gC variants used are expressed by the authors in insect cells. gC is glycosylated during human cell infection. Do the authors know if the glycosylation is similar in their insect cells vs mammalian-expressed gC? The binding data would be stronger if gC expressed by mammalian or optimally human cells was tested. The authors present size exclusion and gel data that the gC fragments look like monomers, but some HHV glycoproteins have a tendency to aggregate when expressed. I am a little worried that gC is aggregating and increasing the net valency of IFN γ after binding IFN γ , similar to a cross-linking mAb, and wondering if the authors can exclude this. Of necessity, the authors use extracellular domain (ECD) truncations of gC, and they narrow down the interaction region somewhat. They then present data with anionic polymers such as heparan that the GAG binding domain of gC may be involved in gC binding to IFN γ . This conclusion would be strengthened by identifying point or deletion mutations in gC that disrupted IFN γ binding. Also related to the protein-protein interaction, have the authors considered using protein folding prediction software for gC, and the known human IFN γ structures, to try to dock or model the purported interaction?

Regarding the binding of gC to IFN γ , can the authors discuss if this is expected to occur in the context of viral infection in vitro or in vivo? gC has a TM domain: is it known if any portion of VZV gC is secreted (as occurs for some HSV glycoprotein fragments for example) in the context of viral infection? Which cell membranes does VZV gC distribute into during infection...does this include the plasma membrane? Is there a situation in which gC could encounter IFN γ , perhaps on the surface of an infected cell, to "deliver" IFN γ to IFN γ R, similar to the kind of presentation thought to occur for IL-15?

The next major body of data concerns treatment of HaCaT cells, a virally immortalized keratinocyte cell line, with IFN γ , gC fragments, or a combination of the two. A set of canonical ISG, such as CXCL9,10,11, as well as ICAM1, were synergistically upregulated by the combination. The conclusions would be strengthened in the truncated gC version that did not bind gC was tested and failed to show this synergy. There is a well-known signaling pathway through IFN γ R to the nucleus involving phosphorylation and movement of transcription factors. Insight could be added by showing changes in these implied steps in the induction of ICAM-1 and the chemokines, especially given the complex pattern of ISG "modulation" noted in which the usual suite of ISGs are not monotonically increased.

Along these lines, please clarify if different time points were studied by bulk RNAseq: given enough time, was the pattern of ISG increase more typical of global enhancement of IFN γ signaling or was the interesting specificity maintained? Concerning these mRNA experiments, primary low-passage human keratinocytes are quite available, and the conclusions would be strengthened if the same mRNA level pattern was noted in such cells, even better if from > 1 donor.

The authors then turn to measuring ICAM1 as the cell surface protein levels by flow cytometry. It is positive that the authors tested multiple immortalized cell lines, but again, data from primary cells such as keratinocytes would be welcome. It is noted that SF8 uses iPSC-derived MF, but the authors do not discuss if MF are involved in the pathogenesis of VZV infection. As a minor point, Cytoflex is mentioned in the paper but required an internet lookup to determine this was a flow cytometer. Data from an isotype control mAb for the anti-ICAM1 and anti-MHCII would be welcome. It would also be helpful to see some flow data presented as dotplots or at least histograms rather than bar graphs as in Fig 3 and SF7. Along these lines, presentation of the protein expression data as MFI or "percent positive" above an explicit cutoff would also be an option, but representative raw data would be best. Anti-ICAM1 staining of an immune cell with high levels of ICAM1 could provide context for the absolute levels observed on the HaCaT and other cells. The iPSC cells in SF8 might be a good candidate for looking at ICAM1 levels on a cell type that is a professional APC and might be expected to have high levels. This is particularly important, and this is one of my major concerns with the paper, because the absolute magnitude of the gC enhancement of ICAM1 expression on most of the cell types is small, about 2 fold (fig 3A, SF7). Discussion of the number of repeat experiments concerning protein level enhancement would be optimal. Rigor would again be added by showing that the truncated version of gC that did not bind IFN γ well also did not lead to the ICAM1 protein enhancement. Similar to the mRNA experiments, the protein experiments would be more physiologically relevant if performed on primary human cells.

The authors then turn to a two-cell system and examine cell-cell binding (Fig. 4). Data are strengthened by inclusion of the short control gC construct and by use of the LFA KO Jurkat. The magnitude of the effect is relatively modest.

Cell cell spread is thought to be the medically important mode of dissemination within host during primary varicella. The first set of experiments, prior to adding a second cell, introduce a somewhat contrived in vitro system in which cells are both IFN γ treated and VZ-eGFP- infected, and cells are divided into GFP^{high} and GFP^{low}. The terms tendency and slightly are used in this part of Results here and appropriately are not called out as significant. As a minor point, the GFP virus used is a deletant in ORF57: is this attenuating in vitro and is it known if it is attenuating in vivo (in the mouse systems)? The SF11D data that are felt to be statistically significant are, again, showing a low magnitude effect. Somewhat contradictory data are shown in the right part of SF11E, in which the more highly infected (GFP brighter) HaCaT cells are ICAM1 low, with a literature citation in Results mentioning actual inhibition of ICAM1 by VZV infection. Do the authors feel that the possible effects of gC on ICAM1 expression are dependent on infection levels? SF11E is not presented with a statistical test, and the number of repeat experiments in SF11 is not stated.

The overall hypothesis concerning VZV dissemination is then addressed in Fig 5 in which a second cell is added on top of VZV-infected cells. A novel set of isogenic mutants is used, one of which has GFP fused to the cytoplasmic tail of gC. Does this alter the trafficking of gC (see query about this issue above)? The phenotype for enhanced Jurkat binding is quite weak (1.2 fold and only when a ratio is used, Fig. 5D). Strength would again be added if primary human epithelial cells were used.

Fig. 6 does use primary human PBMC instead of Jurkat cells as recipient cells of infection, a strength. The title of Fig 6 is not accurate, as panels gA-D use Jurkats which are lymphocytes, while 6E-H use PBMC, which contain many cell types. We see a potentially much stronger effect in 6FGH. It would be optimal to know what actual cell type is getting infected, how many PBMC donors were studied, and again to see dotplots of the GFP intensity in these PBMC instead of just a report of the percent positive. It would be interesting to know if the cells with GFP were monocytes, T cells, B cells, etc.

Minor: Annexin V staining is mentioned in methods, but nowhere else. Please clarify and if relevant give reagents and details.

Minor: please provide documentation that LFA1 KO Jurkats (Fig. 4E) indeed do not express LFA1.

I did not see if RNAseq data were placed in a repository.

I did not see if the novel recombinant VZV strains made were full-length sequenced to determine if they had any adventitious/off target mutations. Dr. Depledge is an expert at sequencing VZV (he has done hundreds) and I think it is reasonable to request this. Optimally sequences would be in Genbank etc.

Reviewer #3 (Remarks to the Author):

This is an interesting paper that demonstrates that VZV glycoprotein C directly binds to IFN γ and alters its biological activity. The gC IFN γ complex induces a smaller subset of genes than IFN γ alone, one of which is ICAM1. When epithelial cells are infected with the VZV virus expressing gC, the cells fuse more readily with T cells or PBMC based on ICAM1-LFA1 interactions, resulting in more efficient viral spread. This is a novel and interesting finding and the authors have taken a number of approaches to verifying their hypothesis. There are a few points that need to be considered.

1. for their studies, the authors use recombinant IFN γ and being recombinant it is not glycosylated. Would glycosylated IFN γ change the results?
2. Is the IFN γ -gC complex retained on the cell surface longer than IFN γ alone?
3. Does gC get internalized?
4. gC induces ICAM1 in macrophages without IFN γ . Did the gC treatment induce Type 1 IFN expression?
5. The authors hypothesize that gC binds some type of receptor albeit unknown. Did gC treatment induce any level of STAT1 phosphorylation or does the IFN γ -gC complex result in prolonged STAT1 phosphorylation? (note that this reviewer appreciates that this question goes to mechanism of the effect and that question is a bit outside the focus of the paper).
6. One might hypothesize that gC primes the chromatin of the 40 specific genes induced by the complex. Is there anything common regarding these genes (unique (e.g. unique GAS elements) that might explain their specific expression?
7. Did heparitinase treatment of cells, eliminate the effect of gC?

Reviewer #4 (Remarks to the Author):

The article "Viral modulation of type II interferon increases T cell adhesion and virus spread" by Jürgens et al. is a well-written and well-designed study that makes significant contributions to our understanding of how VZV glycoprotein C (gC) modulates the activity of interferon-gamma (IFN- γ) and virus spread. The main findings of the study are as follows:

- VZV glycoprotein C (gC) can bind to IFN- γ and increase the expression of a subset of IFN-stimulated genes (ISGs), including intercellular adhesion molecule 1 (ICAM1).
- The upregulation of ICAM1 by gC leads to increased T cell adhesion to infected cells, which facilitates virus spread.
- The presence of gC during infection also increased VZV spread from epithelial cells to peripheral blood mononuclear cells.
- gC does not induce a general enhancement of IFN- γ -stimulated genes but rather increases the expression of a subset of specific genes, especially those involved in chemokine-mediated migration.

- The most striking finding was the upregulation of ICAM1, a gene that is involved in T-cell migration. The effect size of ICAM1 was 26-fold higher in the presence of IFN- γ when gC was also present.

- The findings of this study have implications for our understanding of how VZV evades the immune system and spreads throughout the body. They also suggest that gC could be a potential target for the development of new treatments for VZV infection.

The experiments conducted in the study provide strong evidence to support the authors' findings. The experiments include surface plasmon resonance (SPR) binding screening with human interferons (IFNs), transcriptome analysis of HaCaT cells treated with gC and IFN- γ , confirmation using an antibody neutralizing IFN- γ receptor, investigation of gC interaction with IFN- γ through glycosaminoglycan (GAG)-binding regions, adhesion assays with HaCaT and Jurkat cells, determination of the gC region responsible for increased adhesion, multicycle kinetic experiments by SPR, analysis of ICAM1 levels during VZV infection, and an adhesion experiment to assess VZV spread to T cells. The analytical approach used in the study is strong and comprehensive, and the results from the statistical tests are valid and support the findings of the study.

However, there are a few limitations to the study. First, the authors do not provide the RNAseq analysis results, count-table, or differential expression (DEG) analysis (DEGseq2). This information is important for other researchers to be able to reproduce the study and to conduct comparative studies. I would recommend that the authors include this information in the manuscript.

Second, the study does not include donors' gender information. Finally, the manuscript does not include a graphical abstract. This would be a helpful way to explain the mechanism of infection by VZV glycoprotein C (gC) and to summarize the key findings of the study.

Major issues:

- The authors should perform functional enrichment analysis to identify the biological pathways that are involved in the effects of gC on T cell adhesion and virus spread. This would provide additional insights into the mechanism by which gC modulates the activity of IFN- γ .

- The authors should include the results of the Differential gene expression data in addition to the raw data. This would allow other researchers to reproduce the study and conduct comparative studies.

Minor issues:

- Line 97: The term "HSV" is not defined in the manuscript. It would be helpful to define this term early on in the manuscript.

- Line 170: The sentence "Dendrograms showed the same treatment conditions clustered together, indicating the reproducibility of the results" is not accurate. Clustering of samples in a dendrogram can indicate reproducibility, but it is not a guarantee. Other factors can affect the clustering of samples, such as the quality of the data and the choice of the clustering algorithm. The authors should rewrite this sentence to make it more accurate.

- Line 828: The sentence "Rstudio and Excel were used to sort for interesting genes based on adjusted P values and fold changes" is not accurate. Rstudio is just an integrated development environment (IDE) tool. The authors should replace Rstudio with R (R version).

- The manuscript does not cite the source of the DESeq2 software. The authors should include a citation for this software.

Overall, the article "Viral modulation of type II interferon increases T cell adhesion and virus spread" by Jürgens et al. is a well-written and well-designed study that makes significant contributions to our understanding of how VZV glycoprotein C (gC) modulates the activity of interferon-gamma (IFN- γ) and virus spread. However, there are a few limitations to the study, and the authors should address these aforementioned limitations in the revised manuscript.

Dear reviewers,

We would like to express our gratitude for taking the time to review our manuscript and making constructive criticisms, which we have taken into serious consideration. Based on your recommendations, we performed new experiments and implemented several changes in the text, figures and tables that have improved the manuscript. We apologise for taking more time than initially expected to respond to your comments and concerns. This was mainly due to personnel and structural changes that took place in our laboratory in the last few months. For instance, some of the key scientists involved in the investigation left the lab to embark in new professional challenges and we had to train new members to perform critical experiments. Despite this, we believe that we now submit a clearly improved version of the manuscript.

REVIEWER COMMENTS

Reviewer #2 (Remarks to the Author):

For Authors

The author team is composed of excellent investigators from diverse disciplines who have come together with an integrated body of work. The virus studied, VZV, is medically significant despite the (limited) availability of good vaccines. The cell types studied, or at least their in vivo counterparts, specifically epithelial cells and lymphocytes, are highly relevant to the pathophysiology of primary VZV infection. The work builds on prior, related work by this team, for

example <https://journals.plos.org/plospathogens/article?id=10.1371/journal.ppat.1006346>

High level the paper first identifies a novel viral protein-host protein binding interaction in vitro, second presents in vitro discovery data from bulk RNAseq datasets of the possible impact of this binding interaction, third present one-cell protein level data to correlate with the mRNA data, and finally present two-cell data which aims to back up the functional significance of the first three steps. The overall model is actually quite novel: rather than expressing a simple IFN-sequestering protein as an immune evasin, as some poxviruses do, the authors propose that that a viral protein actually binds and increases the biological activity of IFN γ to in turn influence cells to upregulate an adhesion molecule that is an ISG (ICAM1) and thus favor cell to cell transmission of VZV from one cell to another. If validated, this would be quite an original finding. There is no simple in vivo model for VZV infection, such that in vitro studies may be quite valuable before progressing to the artificial immune deficient murine models or attempting to study the issues addressed in NHP with SVV, a homolog of VZV. However, there are significant limitations with each of the major findings which call into question how well

the overall thesis of the work is supported by the data.

We thank the reviewer for highlighting the findings, novelty and potential relevance of our manuscript and for pointing out the limitations. We have addressed the comments below.

The first data presented concerns gC binding to IFN γ . The gC variants used are expressed by the authors in insect cells. gC is glycosylated during human cell infection. Do the authors know if the glycosylation is similar in their insect cells vs mammalian-expressed gC? The binding data would be stronger if gC expressed by mammalian or optimally human cells was tested.

We agree with the reviewer that glycosylation could play a relevant role in the interaction between gC and IFN γ . Therefore, we expressed gC (residues S147-V531) in human HEK ExPi293F cells (ThermoFisher Scientific) and purchased recombinant IFN γ expressed in HEK293 cells (https://www.rndsystems.com/products/recombinant-human-ifn-gamma-hek293-expressed-protein-cf_10067-if). Expression and purification of gC in HEK ExPi293F cells (Suppl. Fig. 1i, new experiment) was not as efficient as in insect cells and the yield was not high enough to perform all experiments that we would have wished for. Therefore, we decided to focus on the binding and functional experiments. Unfortunately, the concentration of the protein preparation was quite low and this resulted in the use of more volume than required for immobilization, reducing the amount of acetate buffer that was needed for amine coupling. This resulted in very low immobilised protein. We then tried to perform a buffer exchange to have gC in the correct buffer, but we lost protein in the process and the immobilisation was not efficient. Therefore, we could not immobilise enough protein on a CM5 chip to perform binding experiments. We decided then to perform functional experiments as suggested by this reviewer below. The results indicate that gC expressed in HEK ExPi293F also modified IFN γ activity as did the protein expressed in insect cells (Suppl. Fig. 7, new experiment).

Other experiments suggest that gC binding to IFN γ is required to modify IFN γ activity (for instance, the truncated gC version that bound IFN γ through a transient interaction (gC-Y419-S523) did not enhance ICAM1 protein expression, Jurkat T cell adhesion (Figure 3A) nor CXCL9, CXCL10 and CXCL11 gene expression as efficiently as gC-S147-V531 (Suppl. Fig. 6, new experiment). Therefore, we conclude that gC expressed in HEK ExPi293F cells binds IFN γ stably enough to increase its activity.

The authors present size exclusion and gel data that the gC fragments look like monomers, but some HHV glycoproteins have a tendency to aggregate when expressed. I am a little worried that gC is aggregating and increasing the net valency of IFN γ after binding IFN γ , similar to a cross-linking mAb, and wondering if the authors can exclude this.

It is very well possible that gC interacts with itself, forming high order oligomers. We would prefer to employ this term rather than "aggregate". It could be that gC oligomers increase the net valency after binding IFN γ , and this could be part of the mechanism of action. However, this increase in valency would probably induce a general increase in IFN γ signalling but it may not explain why it increases some ISGs but not others. We have included a short text discussing this (lines 575-580). To determine whether this happens

and whether it plays a role in gC function will be investigated in a follow up study to determine the mechanism of action of gC.

Of necessity, the authors use extracellular domain (ECD) truncations of gC, and they narrow down the interaction region somewhat. They then present data with anionic polymers such as heparan that the GAG binding domain of gC may be involved in gC binding to IFN γ . This conclusion would be strengthened by identifying point or deletion mutations in gC that disrupted IFN γ binding.

We apologise because this section of the manuscript was not clear in the original submission. We did not show that the GAG-binding domain of gC is involved in the interaction with IFN γ . We showed that the GAG-binding domain of IFN γ is involved in the interaction with gC.

We published previously that residues 24 to 151 of gC are responsible for GAG binding, while residues 140 to 531 do not interact with GAGs (Gonzalez-Motos et al., 2017, PLoS Pathogens, doi: 10.1371/journal.ppat.1006346). The construct used here to determine the role of GAGs in gC-IFN γ interaction (gC-S147-V531) does not bind GAGs.

gC did not inhibit IFN γ activity, and thereby it did not inhibit the interaction of IFN γ with its receptor. So, we hypothesised that gC could interact with the GAG-binding site of IFN γ . Therefore, the experiments performed were designed to determine this. Our data suggest that the GAG-binding domain of IFN γ is involved in interaction with gC and not that the GAG-binding domain of gC is involved in binding IFN γ . We have explained this better now and changed the title of this section to indicate the main conclusion (lines 303-315).

Also related to the protein-protein interaction, have the authors considered using protein folding prediction software for gC, and the known human IFN γ structures, to try to dock or model the purported interaction?

We thank the reviewer for this suggestion. We have indeed used the current Alphafold multimer to predict an equimolar complex of gC and IFN γ . A scoring of the obtained models revealed high PAE scores and pLDDT values above 80% only for approximately 30-50% of residues, suggesting a low confidence for the obtained models. Combined with the fact that the gC/IFN γ interfaces in these models did not superpose well, these indicators suggested that modelling of this interaction does not provide any useful structural insights and we therefore omitted this analysis from the manuscript for clarity reasons. We are currently working on obtaining the crystal structure of gC-IFN γ complex.

Regarding the binding of gC to IFN γ , can the authors discuss if this is expected to occur in the context of viral infection in vitro or in vivo? gC has a TM domain: is it known if any portion of VZV gC is secreted (as occurs for some HSV glycoprotein fragments for example) in the context of viral infection?

Which cell membranes does VZV gC distribute into during infection...does this include the plasma membrane?

These are also interesting questions. It is not known whether gC is secreted, as HSV gC that is released following alternative splicing (Sedlackova et al., 2008 J Virol,

<https://doi.org/10.1128/JVI.00388-08>) or HSV-2 glycoprotein G that is proteolytically cleaved (Kropp et al., 2020 *J Virol* <https://doi.org/10.1128/JVI.01370-20>; Martinez-Martin et al., *JGV*, <https://doi.org/10.1099/jgv.0.000616>; Balachandran et al., 1985, *J Virol*, <https://doi.org/10.1128/JVI.54.3.825-832.1985>; Su et al., 1987 *J Virol* <https://doi.org/10.1128/JVI.61.5.1735-1737.1987>).

However, we previously showed that gC is present in purified virions and that it enhances chemokine activity in this setting (Gonzalez-Motos et al., 2017, *PLoS Pathogens*, doi: 10.1371/journal.ppat.1006346), indicating that it localizes to the viral envelope. Our immunofluorescence results (Supp Fig 16, new experiment) indicate that gC is also present in the plasma membrane. Finally, we have preliminary data suggesting that gC is released in vesicles produced by VZV infected cells (part of another research project). Therefore, we hypothesise that gC can interact with IFN γ either on the virion or at the plasma membrane. We have included a paragraph in the discussion referring to this point (lines 557-560).

Is there a situation in which gC could encounter IFN γ , perhaps on the surface of an infected cell, to "deliver" IFN γ to IFN γ R, similar to the kind of presentation thought to occur for IL-15?

Based on the localization of gC, we hypothesise that it could bind IFN γ on the surface of an infected cell (and maybe also on the virion) and this could lead to interaction with the IFN γ R in cis or in trans. As gC is expressed late during infection, its impact on IFN γ activity will occur when virions are being produced, facilitating then T cell adhesion and virus spread at a time when VZV virions could infect T cells. We have included a short text referring to this in lines 549-551.

The next major body of data concerns treatment of HaCaT cells, a virally immortalized keratinocyte cell line, with IFN γ , gC fragments, or a combination of the two. A set of canonical ISG, such as CXCL9,10,11, as well as ICAM1, were synergistically upregulated by the combination. The conclusions would be strengthened in the truncated gC version that did not bind gC was tested and failed to show this synergy.

We already showed in the original submission that the truncated gC version that bound IFN γ through a transient interaction (gC-Y419-S523) did not enhance ICAM1 protein expression nor Jurkat T cell adhesion (Figure 3A). Since the increased protein levels observed with gC-S147-V531 correlated with enhanced mRNA levels (Figure 3 and Supp. Figure 6), we assumed that gC-Y419-S523 did not increase ICAM1 transcription. However, we agree with the reviewer that it would be better to expand this observation to other ISGs. Therefore, we have performed the suggested experiment and shown that gC-Y419-S523 did not increase the expression of ICAM1, CXCL9, CXCL10 and CXCL11, while gC-S147-V531 did (Suppl. Fig. 6, new experiment).

There is a well-known signaling pathway through IFN γ R to the nucleus involving phosphorylation and movement of transcription factors. Insight could be added by showing changes in these implied steps in the induction of ICAM-1 and the chemokines, especially given the complex pattern of ISG "modulation" noted in which the usual suite of ISGs are not monotonically increased.

We have addressed whether gC alone or together with IFN γ modified STAT1 phosphorylation on tyrosine (Y) 701, required for its translocation to the nucleus and its transcriptional activity. Our results show that gC enhances STAT1 phosphorylation on Y701 and its translocation to the nucleus only in the presence of IFN γ (Figure 3f, g and Supp. Fig. 10 and 11, new experiments). It is possible that gC also modifies the phosphorylation and movement of other transcription factors and/or the expression/phosphorylation/activity of other proteins involved in IFN γ signalling (e.g., other STATs, NF κ B, erk, PI3K, C/EBP, etc). Determining the impact of gC alone and together with IFN γ on IFN γ signalling is beyond the scope of this manuscript and will be further addressed in a follow up project investigating the mechanism of action of gC.

Along these lines, please clarify if different time points were studied by bulk RNAseq: given enough time, was the pattern of ISG increase more typical of global enhancement of IFN γ signaling or was the interesting specificity maintained? Concerning these mRNA experiments, primary low-passage human keratinocytes are quite available, and the conclusions would be strengthened if the same mRNA level pattern was noted in such cells, even better if from > 1 donor.

We only tested 1 time point for bulk RNAseq due to funding limitations. However, we investigated the impact of gC on the expression of specific genes at different time points using RT-qPCR (Supp Figures 5, 7 (new experiment) and 9). The increase in ICAM1 gene expression mediated by gC increased slightly from 4 to 10 hours post-incubation. It will be interesting to determine the impact of gC alone and together with IFN γ on the transcriptome of primary normal human epithelial keratinocytes (NHEK) at different times post-incubation. We will perform these experiments in a follow up project, after securing funding.

The authors then turn to measuring ICAM1 as the cell surface protein levels by flow cytometry. It is positive that the authors tested multiple immortalized cell lines, but again, data from primary cells such as keratinocytes would be welcome.

We have followed the reviewer's suggestion and performed such experiment. Our results clearly show that gC-S147-V531 increased the amount of ICAM1 (about 2.5-fold) at the plasma membrane of primary NHEK in the presence of IFN γ (Figure 3C, new experiment).

It is noted that SF8 uses iPSC-derived MF, but the authors do not discuss if MF are involved in the pathogenesis of VZV infection.

We thank the reviewer for pointing this out. Macrophages are involved in VZV pathogenesis. Initial attempts to infect human monocytes suggested that these cells were non-permissive for VZV infection (Arbeit et al., 1982, Intervirology, <https://doi.org/10.1159/000149304>; Baba et al., 1983, Microbiol Immunol, <https://doi.org/10.1111/j.1348-0421.1983.tb00642.x>). However, a more recent report showed that VZV productively infects monocytes, reduces their viability and longevity and impairs their differentiation into macrophages (Kennedy et al., 2019 JV, <https://doi.org/10.1128/JVI.01887-18>). Interestingly, VZV productively infects macrophages with high efficiency (Arbeit et al., 1982, Intervirology, <https://doi.org/10.1159/000149304>; Kennedy et al., 2019 JV, <https://doi.org/10.1128/JVI.01887-18>). We have included a short

text mentioning the relevance of macrophages in the pathogenesis of VZV infection: "VZV productively infects macrophages, inducing IL-6 expression through Toll-like receptor 2 signalling, potentially playing a role in the inflammatory response to VZV and in pathogenesis (see lines 286-288).

As a minor point, Cytoflex is mentioned in the paper but required an internet lookup to determine this was a flow cytometer.

We apologise for not having clarified this. We have modified the text (see line 801).

Data from an isotype control mAb for the anti-ICAM1 and anti-MHCII would be welcome.

We have included an isotype control in the new experiments comparing ICAM1 levels in HaCaT and iPSC-derived macrophages (Supp. Figure 12a, new experiment) and in the experiments addressing the effect of gC plus IFN γ on ICAM1 and MHCII protein level at the plasma membrane of NHEK. In both cases, the isotype control did not produce any signal. The effect of gC plus IFN γ on ICAM1 expression is shown in Figure 3c (new experiment). The effect on MHCII expression in NHEK was very low and we did not follow it up.

It would also be helpful to see some flow data presented as dotplots or at least histograms rather than bar graphs as in Fig 3 and SF7. Along these lines, presentation of the protein expression data as MFI or "percent positive" above an explicit cutoff would also be an option, but representative raw data would be best.

We have included the requested dotplots. Nat Comm formatting rules indicate that all data of one figure must fit in an A4 page. Due to space limitations, we have included the dotplots and histograms in Supplementary Information 2.

Anti-ICAM1 staining of an immune cell with high levels of ICAM1 could provide context for the absolute levels observed on the HaCaT and other cells. The iPSC cells in SF8 might be a good candidate for looking at ICAM1 levels on a cell type that is a professional APC and might be expected to have high levels. This is particularly important, and this is one of my major concerns with the paper, because the absolute magnitude of the gC enhancement of ICAM1 expression on most of the cell types is small, about 2 fold (fig 3A, SF7).

We had already shown that gC-S147-V531 increased ICAM1 levels in iPSC-derived macrophages. We have compared now the level of ICAM1 on these cells and HaCaT in a new experiment performed in parallel. Our results show that gC-S147-V531 increases the effect of IFN γ on ICAM1 levels in both cell types. However, the basal level of ICAM1 is much higher in macrophages than HaCaT, as suggested by the reviewer (Supplementary Fig. 12a, new experiment).

An enhancement of 2-fold (approx.) may seem low, but it can have a relevant impact as we show in cell adhesion as well as in virus spread.

Discussion of the number of repeat experiments concerning protein level enhancement would be optimal.

We apologise because this information was missing for certain experiments. We have included the number of repeats in the figure legends.

Rigor would again be added by showing that the truncated version of gC that did not bind IFN γ well also did not lead to the ICAM1 protein enhancement. Similar to the mRNA experiments, the protein experiments would be more physiologically relevant if performed on primary human cells.

We already showed in Figure 3a of the original submission that gC-Y419-S523 did not enhance ICAM1 protein level in HaCaT cells. We have now addressed the reviewer's comment with primary NHEK (Figure 3c, new experiment). Our results show that while gC-S147-V531 (that bound IFN γ well) enhanced ICAM1 protein level on the plasma membrane of NHEK, only in the presence of IFN γ , gC-Y419-S523 (a protein that did not bind IFN γ well) did not.

The authors then turn to a two-cell system and examine cell-cell binding (Fig. 4). Data are strengthened by inclusion of the short control gC construct and by use of the LFA KO Jurkat. The magnitude of the effect is relatively modest.

We did not claim that the effect is very strong. However, an increase of more than 2-fold in cell adhesion may seem modest, but can have a relevant impact on pathogenesis since it can lead to more VZV spread, as our data suggest.

Cell cell spread is thought to be the medically important mode of dissemination within host during primary varicella. The first set of experiments, prior to adding a second cell, introduce a somewhat contrived in vitro system in which cells are both IFN γ treated and VZ-eGFP- infected, and cells are divided into GFP high and GFP low. The terms tendency and slightly are used in this part of Results here and appropriately are not called out as significant. As a minor point, the GFP virus used is a deletant in ORF57: is this attenuating in vitro and is it known if it is attenuating in vivo (in the mouse systems)?

VZV ORF57 is a non-essential gene and viruses lacking this gene are not attenuated in cell culture (Zhang et al., 2010 PLoS Pathogens, doi:10.1371/journal.ppat.1000971; Cox et al., 1998, Virology). There are no reports testing the effect of lack of ORF57 in vivo, probably due to the high human-specific nature of VZV. One could test this in the xenografted SCID mouse model or maybe with guinea pigs, but we do not have any of these two models. However, a virus expressing an ORF57-luciferase fusion protein replicated as well as parental virus in vitro (Lloyd et al., 2022 Viruses, <https://doi.org/10.3390/v14040826>) and was employed to test antivirals in human skin in vitro and in human skin xenografted in SCID-beige and athymic nude mice (Lloyd et al., 2020 JVirol, <https://doi.org/10.1128/JVI.01082-20>).

The SF11D data that are felt to be statistically significant are, again, showing a low magnitude effect. Somewhat contradictory data are shown in the right part of SF11E, in which the more highly infected (GFP brighter) HaCaT cells are ICAM1 low, with a literature citation in Results mentioning actual inhibition of ICAM1 by VZV infection. Do the authors feel that the possible effects of gC on ICAM1 expression are dependent on

infection levels? SF11E is not presented with a statistical test, and the number of repeat experiments in SF11 is not stated.

It is well possible that the effect of gC on ICAM1 levels are dependent on the infection and gC expression levels. However, we do not have any data supporting this hypothesis. The results shown correspond to three independent biological experiments. This is indicated in the figure legend (now Supp. Fig. 15).

The overall hypothesis concerning VZV dissemination is then addressed in Fig 5 in which a second cell is added on top of VZV-infected cells. A novel set of isogenic mutants is used, one of which has GFP fused to the cytoplasmic tail of gC. Does this alter the trafficking of gC (see query about this issue above)? The phenotype for enhanced Jurkat binding is quite weak (1.2 fold and only when a ratio is used, Fig. 5D). Strength would again be added if primary human epithelial cells were used.

We have investigated whether the presence of GFP modifies the subcellular localization of gC by performing immunofluorescence (Supp. Fig. 16b, new experiment). Our results suggest that addition of GFP to the cytoplasmic tail of gC does not modify its localization. We also tried to repeat the adhesion experiment with NHEK. However, we had problems efficiently infecting these cells with our VZV reporter strains. Optimisation of these infection experiments requires more work and time and will be performed in a future project as indicated in lines 524-526.

Fig. 6 does use primary human PBMC instead of Jurkat cells as recipient cells of infection, a strength. The title of Fig 6 is not accurate, as panels gA-D use Jurkats which are lymphocytes, while 6E-H use PBMC, which contain many cell types.

We apologise for this mistake. We have now written "VZV-gC-GFP spreads more efficiently than VZV-ΔgC-GFP from HaCaT cells to Jurkat cells and PBMCs than VZV-ΔgC-GFP" (lines 1352-1353).

We see a potentially much stronger effect in 6FGH. It would be optimal to know what actual cell type is getting infected, how many PBMC donors were studied, and again to see dotplots of the GFP intensity in these PBMC instead of just a report of the percent positive. It would be interesting to know if the cells with GFP were monocytes, T cells, B cells, etc.

We agree that it would be interesting to know which cell type is infected. We plan to do this in a follow up study in which we will also investigate the impact of VZV infection on leukocyte function. We have included a sentence in lines 524-526 indicating this. We have added the dotplots of the GFP intensity (Supplementary Information 2), and indicated the number of donors studied (two donors; see line 471).

Minor: Annexin V staining is mentioned in methods, but nowhere else. Please clarify and if relevant give reagents and details.

We apologise for the lack of information on annexin V. We employed annexin V to quantify apoptotic cells. We have included this information in lines 794-795. We employed Annexin V in the initial experiments shown in Figure 3a,b, because IFN γ could induce this

biological process. Since we did not observe apoptotic cells due to IFN γ , gC or both, we excluded this staining in the following experiments.

Minor: please provide documentation that LFA1 KO Jurkats (Fig. 4E) indeed to do not express LFA1.

This cell line was previously generated and characterized (Hain et al., 2018, doi: 10.1038/s41598-018-21344-7). We have confirmed the lack of LFA1 expression by FACS (Figure 4e, new experiment).

I did not see if RNAseq data were placed in a repository.

We apologise for this. We had uploaded the RNAseq data in the Gene Expression Omnibus (GEO) repository, and included this information in line 831 of the original submission. However, due to unknown reasons, the data was not available. We have solved this problem. The RNAseq data can be accessed in European Nucleotide Archive repository, (accession number PRJEB61951). We have also included additional data such as the raw count table and the DeSeq2 analyses as Data Source 1.

I did not see if the novel recombinant VZV strains made were full-length sequenced to determine if they had any adventitious/off target mutations. Dr. Depledge is an expert at sequencing VZV (he has done hundreds) and I think it is reasonable to request this. Optimally sequences would be in Genbank etc.

We apologise for this. We have sequenced the viruses following their reconstitution in eukaryotic cells and uploaded the sequences in Genbank. We performed a hybrid approach using Illumina and Oxford Nanopore (see details in Supp. Materials and Methods). We have included the accession numbers in the "Data availability" section of the manuscript. pOka_Delta_57_GFP_09_2023, Acc. N. PP378487; pOka_Delta_gC_GFP_09_2023, Acc. N. PP378488; pOka_gC_GFP_09_2023, Acc. N. PP378489

There were no undesired modifications in the sequences of pOka-gC-GFP and pOka-DelgC-GFP. Sequencing of pOka-Del57-GFP revealed that there was no stop codon in ORF58. We do not expect this to affect our results since VZV ORF58 is not required for replication in cell culture (Yoshii et al., 2008 Virol J, doi:10.1186/1743-422X-5-54).

Reviewer #3 (Remarks to the Author):

This is an interesting paper that demonstrates that VZV glycoprotein C directly binds to IFN γ and alters its biological activity. The gC IFN γ complex induces a smaller subset of genes than IFN γ alone, one of which is ICAM1. When epithelial cells are infected with the VZV virus expressing gC, the cells fuse more readily with T cells or PBMC based on ICAM1-LFA1 interactions, resulting in more efficient viral spread. This is a novel and interesting finding and the authors have taken a number of approaches to verifying their hypothesis. There are a few points that need to be considered.

We thank the reviewer for the positive comments and for providing suggestions to improve the manuscript.

1. for their studies, the authors use recombinant IFN γ and being recombinant it is not glycosylated. Would glycosylated IFN γ change the results?

We agree with the reviewer that glycosylation of IFN γ could be important. The recombinant IFN γ employed in the original submission was expressed in bacteria and therefore lacked glycosylation. To address the reviewer's comment, we purchased IFN γ from R&D that is expressed in human HEK293 cells and is therefore glycosylated (https://www.rndsystems.com/products/recombinant-human-ifn-gamma-hek293-expressed-protein-cf_10067-if). We termed this IFN γ as m (for mammalian) IFN γ . Similarly, the gC constructs employed in the original study were expressed in insect cells that impose different glycosylation than mammalian cells. Therefore, we tested whether the use of IFN γ and gC produced in mammalian cells had a similar effect as the ones produced in other systems. The combination of gC-S147-V531 and IFN γ (both expressed in human cells) increased expression of ICAM1, CXCL9, CXCL10 and CXCL11 in HaCaT (Supp. Fig. 7, new experiment). These results are similar to those obtained with IFN γ expressed in bacteria, suggesting that glycosylation of IFN γ does not influence the results reported here.

We also determined the effect of gC-S147-V531 on the translocation of p-STAT1 (see also point 5 below, Figure 3g and Supp. Fig. 10 and 11, new experiment) using IFN γ expressed in insect cells and HEK293 cells. gC-S147-V531 increased p-STAT1 nuclear translocation in both experimental conditions, suggesting again that glycosylation of IFN γ does not influence the results reported here.

2. Is the IFN γ -gC complex retained on the cell surface longer than IFN γ alone?

Our data indicate that the gC-IFN γ complex acts through the IFN γ R. Therefore, we tried to answer this interesting question by detecting IFN γ R by FACS at different time points following incubation at 4°C and 37°C with gC, IFN γ and both. Unfortunately, we could not reach any conclusion. To determine whether IFN γ -gC complex remains longer than IFN γ alone will help dissecting the mechanism of action of gC. However, we feel that this is currently out of the scope of this manuscript, since there are several potential ways for gC to modulate IFN γ activity. To answer this question requires the establishment of new methods and, therefore, we are currently writing a grant application for this.

3. Does gC get internalized?

We tried to address this question by incubating cells with gC fused to an mNeon tag (gC-mNeon) and detecting the signal by fluorescence microscopy. However, for unknown reasons, this gC prep did not bind IFN γ efficiently. We also could not observe green fluorescence when incubating HaCaT cells with gC-mNeon (irrespective of the presence of IFN γ), suggesting that it did not interact with the plasma membrane. It is possible that the mNeon tag causes steric hindrance that impedes the gC-IFN γ interaction. Therefore, we could not address whether gC gets internalized. To investigate this interesting question, we will need to generate and validate other gC constructs and this will be expensive and time consuming. We will address this question in a future project investigating the mechanism of action of gC.

4. gC induces ICAM1 in macrophages without IFN γ . Did the gC treatment induce Type 1 IFN expression?

This is an interesting question. We detected INF β expression by RT-qPCR. Incubation with gC did not INF β expression. We have not included this negative result in the current manuscript. However, if the reviewer considers that its publication is required, we will include it.

5. The authors hypothesize that gC binds some type of receptor albeit unknown. Did gC treatment induce any level of STAT1 phosphorylation or does the IFN γ -gC complex result in prolonged STAT1 phosphorylation? (note that this reviewer appreciates that this question goes to mechanism of the effect and that question is a bit outside the focus of the paper).

We agree with the reviewer that this is a very interesting question that deserves an extensive investigation. We are working on obtaining funding to investigate the mechanism of action of gC and whether this involves the activity of another receptor. However, we have followed the reviewer's suggestion and addressed whether gC modifies STAT1 phosphorylation at Y701 and subsequent nuclear translocation. Our results show that gC increases p-STAT1 and its nuclear translocation in the presence of IFN γ (Figure 3f, g and Supp. Fig. 10 and 11, new experiments). gC alone did not induce detectable levels of p-STAT1 nor increased p-STAT1 nuclear translocation. The effects were noticeable at both 10 and 30 minutes post-incubation. These results suggest that gC may not interact with another receptor that leads to activation of the JAK/STAT pathway.

6. One might hypothesize that gC primes the chromatin of the 40 specific genes induced by the complex. Is there anything common regarding these genes (unique (e.g. unique GAS elements) that might explain their specific expression?

If we understand this question correctly, the reviewer would like to know whether the promoters of these genes contain any common sequences that would lead to binding of specific transcription factors (or a combination of those). We checked the presence of binding sequences for different transcription factors/complexes, using different available software such as <https://jaspar.genereg.net/>; <https://www.expasy.org/search/Transcription%20factor%20binding%20site%20prediction?type=keyword>; <https://genexplain.com/transfac-features-and-packages/tfbs-prediction-from-portal-interface/>.

However, we did not find a specific or common "signature" with this approach (lines 578-580). The only potential interesting signature is that several of the genes contain many NF κ B-binding sites. For instance, ICAM1 contains 85 and IL4I1, 69. Therefore, we think that gC could promote IFN γ -mediated expression of ICAM1 through NF κ B. We will follow this up in a project investigating the mechanism of action of gC.

7. Did heparitinase treatment of cells, eliminate the effect of gC?

We will address this question in the context of the investigation of gC mechanism of action. However, we hypothesise that lack of GAGs will not affect gC function. This is based on the following: the gC constructs employed here that increase IFN γ activity do not bind GAGs,

since they lack residues 24 to 151, responsible for GAG-binding (Gonzalez-Motos et al., 2017 PLoS Pathogens, doi: 10.1371/journal.ppat.1006346). Our competition experiments with GAGs suggest that gC interacts with IFN γ through the GAG-binding site of this cytokine. Therefore, this domain of IFN γ should not be available to interact with GAGs.

Reviewer #4 (Remarks to the Author):

The article "Viral modulation of type II interferon increases T cell adhesion and virus spread" by Jürgens et al. is a well-written and well-designed study that makes significant contributions to our understanding of how VZV glycoprotein C (gC) modulates the activity of interferon-gamma (IFN- γ) and virus spread. The main findings of the study are as follows:

- VZV glycoprotein C (gC) can bind to IFN- γ and increase the expression of a subset of IFN-stimulated genes (ISGs), including intercellular adhesion molecule 1 (ICAM1).
- The upregulation of ICAM1 by gC leads to increased T cell adhesion to infected cells, which facilitates virus spread.
- The presence of gC during infection also increased VZV spread from epithelial cells to peripheral blood mononuclear cells.
- gC does not induce a general enhancement of IFN- γ -stimulated genes but rather increases the expression of a subset of specific genes, especially those involved in chemokine-mediated migration.
- The most striking finding was the upregulation of ICAM1, a gene that is involved in T-cell migration. The effect size of ICAM1 was 26-fold higher in the presence of IFN- γ when gC was also present.
- The findings of this study have implications for our understanding of how VZV evades the immune system and spreads throughout the body. They also suggest that gC could be a potential target for the development of new treatments for VZV infection.

The experiments conducted in the study provide strong evidence to support the authors' findings. The experiments include surface plasmon resonance (SPR) binding screening with human interferons (IFNs), transcriptome analysis of HaCaT cells treated with gC and IFN- γ , confirmation using an antibody neutralizing IFN- γ receptor, investigation of gC interaction with IFN- γ through glycosaminoglycan (GAG)-binding regions, adhesion assays with HaCaT and Jurkat cells, determination of the gC region responsible for increased adhesion, multicycle kinetic experiments by SPR, analysis of ICAM1 levels during VZV infection, and an adhesion experiment to assess VZV spread to T cells. The analytical approach used in the study is strong and comprehensive, and the results from the statistical tests are valid and support the findings of the study.

However, there are a few limitations to the study. First, the authors do not provide the RNAseq analysis results, count-table, or differential expression (DEG) analysis (DEGseq2). This information is important for other researchers to be able to reproduce the study and to conduct comparative studies. I would recommend that the authors include this information in the manuscript.

We thank the reviewer for highlighting the relevance of our findings and for suggesting modifications that we have implemented.

We apologise for not including the RNAseq analysis results, count-table, or differential expression (DEG) analysis in the original submission. We had uploaded the RNAseq data in the Gene Expression Omnibus (GEO) repository, and included this information in line 831 of the original submission. However, due to unknown reasons, the data was not available. We have solved this problem. The RNAseq data can be accessed through the European Nucleotide Archive repository, (accession number PRJEB61951). We have also included additional data such as the raw count table and the DeSeq2 analyses in Data Source 1.

Second, the study does not include donors' gender information.

We did not include the gender information of blood donors due to ethical reasons. We obtained permission to work with PBMCs from anonymised donors. All donors that provided NHEK were male (NHEK prepared from surgical residuals of foreskin obtained from anonymized children undergoing surgery).

Finally, the manuscript does not include a graphical abstract. This would be a helpful way to explain the mechanism of infection by VZV glycoprotein C (gC) and to summarize the key findings of the study.

We thank the reviewer for this suggestion. However, the Editor informed us that Nature Communications does not include graphical abstracts in its publications

Major issues:

- The authors should perform functional enrichment analysis to identify the biological pathways that are involved in the effects of gC on T cell adhesion and virus spread. This would provide additional insights into the mechanism by which gC modulates the activity of IFN- γ .

We are thankful for this suggestion. We have performed functional enrichment analysis (Figure Supplementary Fig. 4c, new results) supporting that gC modulates expression of proteins involved in chemotaxis (chemokine receptors). Interestingly, gG also modulates the expression of proteins involved in IL-10, IL-13 and IL-4 signalling pathways, which could impact T cell activity. Investigating the effect of gC in these different pathways requires further work.

- The authors should include the results of the Differential gene expression data in addition to the raw data. This would allow other researchers to reproduce the study and conduct comparative studies.

We apologise because this information was not available in the original submission. As indicated above, we have included the data in Data Source 1.

Minor issues:

- Line 97: The term "HSV" is not defined in the manuscript. It would be helpful to define this term early on in the manuscript.

We apologise for this. We have not indicated the meaning of this acronym the first time that it is used (line 93).

- Line 170: The sentence "Dendrograms showed the same treatment conditions clustered together, indicating the reproducibility of the results" is not accurate. Clustering of samples in a dendrogram can indicate reproducibility, but it is not a guarantee. Other factors can affect the clustering of samples, such as the quality of the data and the choice of the clustering algorithm. The authors should rewrite this sentence to make it more accurate.

We have rewritten this sentence. It now reads as "Dendrograms showed that, in line with the PCA plot (Supp Fig. 3a), treatment conditions clustered appropriately (Fig. 2B)" (line 168).

- Line 828: The sentence "Rstudio and Excel were used to sort for interesting genes based on adjusted P values and fold changes" is not accurate. Rstudio is just an integrated development environment (IDE) tool. The authors should replace Rstudio with R (R version).

Thanks for the clarification. We have corrected this. The sentence now reads "R version R-4.1.2 (managed through RStudio version 2021.09.2+382) and Excel were used to sort for interesting genes based on adjusted P values and fold changes (line 761-762).

- The manuscript does not cite the source of the DESeq2 software. The authors should include a citation for this software.

*We have added the following citation: Love, M. I., et al. (2014). "Moderated estimation of fold change and dispersion for RNA-seq data with DESeq2." *Genome Biol* 15(12): 550; DOI: 10.1186/s13059-014-0550-8*

Overall, the article "Viral modulation of type II interferon increases T cell adhesion and virus spread" by Jürgens et al. is a well-written and well-designed study that makes significant contributions to our understanding of how VZV glycoprotein C (gC) modulates the activity of interferon-gamma (IFN- γ) and virus spread. However, there are a few limitations to the study, and the authors should address these aforementioned limitations in the revised manuscript.

We thank the reviewer for the nice and encouraging comments and for pointing out the limitations, which we have addressed.

REVIEWERS' COMMENTS

Reviewer #2 (Remarks to the Author):

NCOMMS-23-21736A

Corresponding Author: Abel Viejo-Borbolla

Title: Viral modulation of type II interferon increases T cell adhesion and virus spread

Authors

The Authors have performed significant additional experiments that respond to many of the critiques raised by myself and the other reviewers. In other cases where additional experiments were requested, the authors have cited financial limitations and plans for future research. Several new authors are added. The Novelty of the proposed pathogenesis mechanism remains high and is an attractive feature. The applicability to what we know about the pathogenesis of VZV also remains reasonable in terms of the cell types investigated.

One improvement is the demonstration that mammalian expressed truncated gC and mammalian IFN γ when admixed seem to have a synergistic activity inducing ISG. We still don't have a crystal structure or co-IP for these molecular species and modeling (Rosetta, AlphaFold etc.) does not dock the molecules well, so the physical binding is based on the BiaCore type measurements. There are some tantalizing data concerning GAGs that may inhibit binding.

The authors have shown that the gC-IFN γ combination does signal through STAT-1 with phosphorylation as expected. The authors also supply underlying flow cytometry dot plot examples to support their bar graphs. Additional methods details and data depositions are provided as requested.

Overall the author team has substantively revised the MS as suggested and I suggest it is acceptable.

Reviewer #3 (Remarks to the Author):

The authors made a significant effort to address my concerns. I have no further comments

Reviewer #4 (Remarks to the Author):

Dear Editor,

I am writing to you regarding my second review of the manuscript titled "Viral modulation of type II interferon increases T cell adhesion and virus spread" by Abel Viejo-Borbolla (Corresponding Author). I previously reviewed this manuscript and provided feedback to the editor.

I am pleased to report that I have carefully reviewed the revised manuscript and found that the authors have addressed all of the concerns I raised in my initial review. They have provided satisfactory responses to my queries and implemented the suggested changes effectively.

In particular, the Transcriptomics data and functional enrichment analysis, which were missing in the previous version, have now been included in the revised manuscript (Data Source 1 and supplementary fig. 4b). This addition significantly strengthens the manuscript by providing a deeper understanding of the underlying biological processes.

Overall, I am satisfied with the revisions made by the author. I recommend its acceptance for publication in "Nature Communications".

Sincerely,

Nilesh Kumar (Ph.D)

Bioinformatician III, Ircp-Biological Data Sciences

University of Alabama at Birmingham, AL, USA